# Mitigating Hidden Confounding by Progressive Confounder Imputation via Large Language Models

## Abstract

Hidden confounding remains a central challenge in estimating treatment effects from observational data, as unobserved variables can lead to biased causal estimates. While recent work has explored the use of large language models (LLMs) for causal inference, most approaches still rely on the unconfoundedness assumption. In this paper, we make the first attempt to mitigate hidden confounding using LLMs. We propose ProCI (Progressive Confounder Imputation), a framework that elicits the world knowledge of LLMs to iteratively generate, impute, and validate hidden confounders. ProCI leverages two key capabilities of LLMs: their strong semantic reasoning ability, which enables the discovery of plausible confounders from both structured and unstructured inputs, and their embedded world knowledge, which supports counterfactual reasoning under latent confounding. To improve robustness, ProCI adopts a distributional reasoning strategy instead of direct value imputation to prevent the collapsed outputs. Extensive experiments demonstrate that ProCI uncovers meaningful confounders and significantly improves treatment effect estimation across various datasets and LLMs.

## 1 Introduction

Estimating treatment effects from observational data is a central problem in causal inference, with widespread applications in healthcare (Prosperi et al., 2020; Grootendorst, 2007), social sciences (Gangl, 2010), and economics (Varian, 2016; Sun et al., 2024). Different from Randomized Controlled Trials (RCTs), the non-random treatment assignments between treatment and control groups in observational studies can lead to imbalanced confounders between groups, which has been recognized as a key contributor to non-causal associations (Pearl, 2009). The bias caused by imbalanced confounders is referred to as confounding bias (Shalit et al., 2017; Wang et al., 2023).

To estimate treatment effects in an unbiased manner from observational data, recent advances have focused on representation-based methods that aim to balance latent distributions between treatment and control groups (Shalit et al., 2017; Wang et al., 2023; Assaad et al., 2021; Yao et al., 2018). More recently, with the emergence of large language models (LLMs) demonstrating remarkable capabilities in reasoning and knowledge understanding, a plenty of work has begun to explore their potential in treatment effect estimation (Ma, 2024; Hüyük et al., 2024; Jin et al., 2023; Han et al., 2024; Imai & Nakamura, 2024). For example, Jin et al. (2023) prompts LLMs using chain-of-thought reasoning to generate natural language rationales that support causal conclusions, while Imai & Nakamura (2024) leverages LLMs to encode high-dimensional text treatments for treatment effect estimation while mitigating confounding.

However, existing methods including those based on LLMs, mostly rely on the *unconfoundedness assumption* (Pearl, 2009), which posits that all confounders affecting both treatment and outcome are observed. This assumption is often unrealistic in real-world settings, where important factors may be unobserved or entirely unrecorded (Ananth & Schisterman, 2018). For example, in healthcare, treatment decisions may be influenced not only by observed clinical features such as age, but also by unobserved factors like socioeconomic status, which can act as confounders if omitted. To address hidden confounding issue, prior work has explored *three main directions*. Sensitivity analysis aims to quantify the impact of hidden confounding by deriving bounds on treatment effects (Robins et al.,

2000; Rosenbaum & Rubin, 1983), but typically relies on fixed and untestable assumptions about the confounding mechanism (Franks et al., 2018; Veitch & Zaveri, 2020). Auxiliary variable methods, including instrumental variables and front-door adjustment, use external information or intermediate pathways to recover unbiased estimates (Li et al., 2023; Fulcher et al., 2017; Shah et al., 2023), but depend on strong structural assumptions that are rarely verifiable (Imbens, 2014; Wu et al., 2022; Bellemare et al., 2024). RCT integration methods combine randomized and observational data to correct for hidden confounding bias (Kallus et al., 2018; Hatt et al., 2022; Wu & Yang, 2022), but the high cost and limited availability of RCTs often restrict their practical utility.

**To fill this gap, we make the first attempt to leverage LLMs to mitigate hidden confounding in treatment effect estimation.** Compared to traditional methods, LLMs offers two key advantages. First, LLMs possess strong semantic reasoning capabilities that allow them to interpret both structured covariates and unstructured textual descriptions, enabling the discovery of plausible confounders that are not explicitly recorded in the data. Second, LLMs embed extensive world knowledge learned from large-scale corpora, which implicitly captures a wide range of latent confounders, allowing them to support counterfactual reasoning even under hidden confounding.

To elicit these capabilities in practice, we propose **ProCI** (Progressive Confounder Imputation), a novel framework that leverages LLMs to iteratively uncover and adjust for hidden confounders. ProCI alternates between two phases: (1) *confounder imputation*, where the LLM is encouraged to generate a plausible missing confounder based on the semantics of the observed variables, and to impute its concrete values; and (2) *unconfoundedness validation*, which builds on the LLM's counterfactual reasoning ability—by prompting the LLM to impute missing potential outcomes, we empirically test whether the generated confounders restore conditional independence between treatment and outcomes. Rather than directly predicting values, which we find empirically leads to collapsed or inconsistent outputs, ProCI adopts a distributional reasoning strategy: it first elicits from the LLM the most plausible distribution type based on its world knowledge, and then infers distribution parameters to generate diverse, realistic samples. This progressive process continues until the generated confounders pass the independence test, indicating that key sources of hidden confounding have been mitigated. By incorporating LLMs into the treatment effect estimation pipeline, ProCI provides a flexible and scalable solution for mitigating hidden confounding in observational studies. Our contributions can be summarized as follows:

• To the best of our knowledge, this is the first work that leverages LLMs to mitigate hidden confounding in treatment effect estimation. This is enabled by eliciting both the semantic signals in observational data and the implicit confounding knowledge embedded in LLMs.

• We propose ProCI, a progressive framework that elicits LLMs to generate and impute plausible confounders by jointly leveraging textual descriptions and structured covariates. The sufficiency of the generated confounders is validated through LLM-based counterfactual prediction and an empirical test of conditional independence.

• Extensive experiments across diverse datasets and multiple LLM architectures demonstrate the effectiveness and generality of our approach, showing improved performance in uncovering hidden confounders and estimating treatment effects.

## 2 PRELIMINARIES

### 2.1 PROBLEM SETUP

We consider the standard setup in binary treatment effect estimation[1]. Let $T \in \{0, 1\}$ denote a binary treatment assignment, where $T = 1$ indicates receiving the treatment and $T = 0$ denotes the control group. Let $Y \in \mathbb{R}$ be the observed outcome, and $X \in \mathcal{X}$ be the observed covariates. The observational dataset consists of i.i.d. samples $\{(x_i, t_i, y_i)\}_{i=1}^{n} \sim \mathcal{D}$. Under the potential outcomes framework (Rubin, 2005), each unit is associated with two potential outcomes: $Y^1$ and $Y^0$, representing the outcomes the unit would receive under treatment and control, respectively. In practice, only one factual outcome $Y$ is observed for each unit, depending on the assigned treatment.

---

[1]While we focus on the binary treatment setting for clarity, the proposed method of this paper can be directly extended to other scenarios, such as multi-valued or continuous treatments.

The individual-level treatment effect is defined as:

$$\tau(x) = \mathbb{E}[Y^1 - Y^0 \mid X = x], \tag{1}$$

which is commonly referred to as the *Conditional Average Treatment Effect* (CATE) (Abrevaya et al., 2015). It quantifies the expected difference in potential outcomes for a given covariate profile $x$, and serves as the primary estimand of interest in individual-level treatment effect estimation.

## 2.2 IDENTIFIABILITY

CATE estimation from observed data requires several identifiability assumptions (Pearl, 2009), including consistency, positivity, SUTVA, and unconfoundedness. A detailed introduction to these assumptions can be found in Appendix B. This paper focuses on the unconfoundedness assumption, which we formally define as follows:

**Assumption 1** (Unconfoundedness). *The treatment assignment is independent of the potential outcomes, conditional on the observed covariates, i.e.,* $(Y^1, Y^0) \perp\!\!\!\perp T \mid X$.

When these assumptions hold, the CATE function in Eq. (1) becomes identifiable as follows:

$$\tau(x) = \mathbb{E}[Y \mid T = 1, X = x] - \mathbb{E}[Y \mid T = 0, X = x], \tag{2}$$

which can be directly estimated from a finite observational dataset $\mathcal{D}$ by learning an estimator $\hat{\tau}(x)$, *e.g.*, using CFRNet or TARNet (Shalit et al., 2017), that approximates the true CATE function $\tau(x)$.

**Hidden Confounding Challenge**. However, in practice, the assumption of unconfoundedness can be easily violated due to limited context (Jesson et al., 2021), where observed covariates $X$ do not capture all relevant confounders that jointly affect treatment and outcome. In particular, treatment assignment may depend on hidden variables $U$ that also influence the outcome $Y$. In such cases, the conditional independence assumption $(Y^1, Y^0) \perp\!\!\!\perp T \mid X$ is no longer valid, and is instead replaced by $(Y^1, Y^0) \not\!\perp\!\!\!\perp T \mid X$, but possibly $(Y^1, Y^0) \perp\!\!\!\perp T \mid (X, U)$ if $U$ were additionally observed. This hidden confounding results in a systematic estimation bias: the learned estimator $\hat{\tau}(x)$, which assumes unconfoundedness given $X$, is biased with respect to the true CATE $\tau(x)$.

# 3 THE PROPOSED PROCI FRAMEWORK

## 3.1 MOTIVATION

Although a range of methods have been proposed to mitigate hidden confounding issue, including auxiliary variables, sensitivity analysis, and data from RCTs, as described in Section 1, these approaches either rely on untestable assumptions or are prohibitively expensive to apply in practice (Ananth & Schisterman, 2018). Motivated by these, we propose a new paradigm for addressing hidden confounding through LLMs. Leveraging their unique capabilities, this paper makes the first attempt to directly mitigate hidden confounding using LLMs, based on two key advantages:

• **Advantage 1: Semantic Utilization Beyond Tabular Data.** Traditional methods typically rely on structured tabular data and predefined variable sets, making it difficult to discover missing confounders that are not explicitly recorded. In contrast, as shown in Figure 1 (A1-1) and (A1-2), LLMs can generate meaningful confounder candidates by interpreting the semantic relationships among treatment, outcome, and observed covariates. This demonstrates the LLM's ability to reason about plausible latent variables using domain-level priors encoded in language. Subsequently, by leveraging the existing tabular data, the LLM can perform unit-level confounder imputation based on its contextual inference capabilities to complete missing data.

• **Advantage 2: Implicit Confounding Awareness via World Knowledge.** Beyond explicit confounder generation, LLMs inherently encode rich world knowledge that implicitly captures causal dependencies and latent confounding factors. As illustrated in Figure 1 (A2), this embedded knowledge allows LLMs to approximate the influence of hidden variables and thus support counterfactual reasoning without requiring direct access to all confounders. Such implicit awareness makes LLMs particularly suited for scenarios where collecting complete causal information is impractical, offering a scalable alternative to traditional approaches. This advantage becomes increasingly valuable as the complexity and dimensionality of real-world data continue to grow.

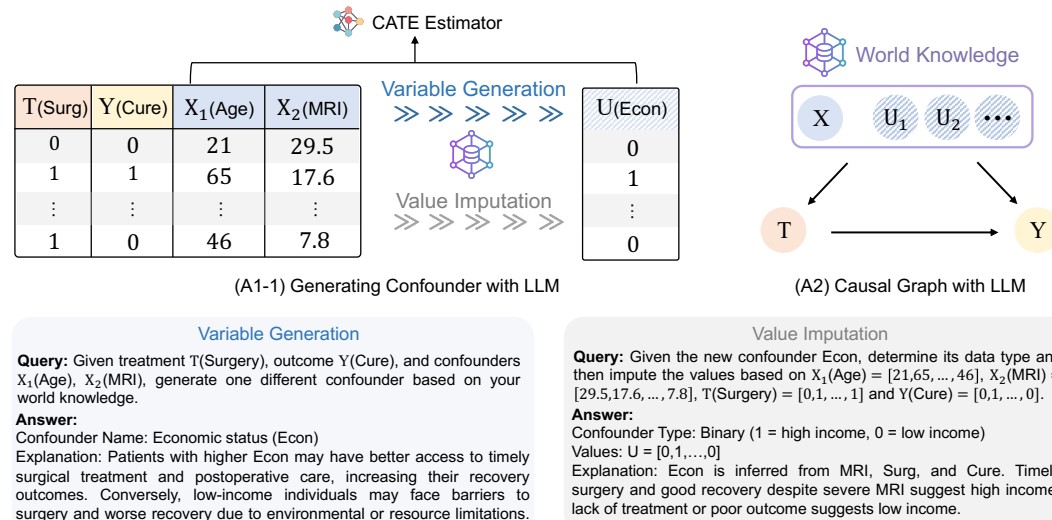

Figure 1: Motivating Illustration. **(A1-1)** The LLM takes structured data to generate a missing confounder and impute its individual-level values for CATE estimation. **(A1-2)** Exemplar queries and responses show how LLMs leverage semantic and world knowledge in variable generation and value imputation. **(A2)** From a causal perspective, LLMs embed latent confounders within their world knowledge, enabling counterfactual reasoning without requiring explicit access to them.

## 3.2 OVERVIEW OF THE PROCI FRAMEWORK

To elicit the above capabilities of LLMs for practical use, we propose ProCI, a framework for mitigating hidden confounding in observational data. As illustrated in Figure 2, ProCI consists of two iterative phases: *confounder imputation* and *unconfoundedness validation*.

**Phase 1: Confounder Imputation.** Given the observed dataset $\{X^{(0)}, T, Y\}$, where $X^{(0)}$ contains covariates, ProCI guides the LLM to generate a plausible confounder $\hat{U}$ based on the semantic relationships among $X^{(k)}$, $T$, and $Y$, along with a textual explanation. It then imputes unit-level values for $\hat{U}$ by determining its distribution type and parameters via LLM-guided reasoning. The imputed confounder is added to the dataset to form $X^{(k+1)}$, where $k$ denotes the iteration step.

**Phase 2: Unconfoundedness Validation.** To assess whether the generated confounder $\hat{U}$ captures sufficient hidden bias, ProCI performs an imputation-based empirical unconfoundedness test. Using the LLM's world knowledge, ProCI first imputes counterfactual outcomes $\hat{Y}^0$ and $\hat{Y}^1$, and then applies a kernel-based conditional independence test (KCIT) to check if $(\hat{Y}^0, \hat{Y}^1) \perp\!\!\!\perp T \mid X^{(k+1)}$. If the test fails, ProCI returns to Phase 1 to generate an additional confounder; if it passes, the framework proceeds to CATE estimation using a standard estimator like TARNet.

By progressively generating and validating confounders, ProCI adaptively constructs a sufficient adjustment set without needing access to ground-truth confounders, enabling more reliable treatment effect estimation in the presence of hidden bias.

## 3.3 CONFOUNDER IMPUTATION VIA LLMS

In this section, we propose to prompt LLMs to generate hidden confounders that are semantically grounded and causally relevant. Imputing a confounder involves two key steps: (1) generating a variable with a meaningful textual description, and (2) imputing its concrete values for each individual in the dataset. We describe these steps in detail below.

**Variable Generation.** Given a dataset $\mathcal{D}_{\text{obs}}$ consisting of observed variables $\{X, T, Y\}$, where $X$ denotes covariates, $T$ the treatment, and $Y$ the outcome, we first design a prompt function $\mathcal{P}_{\text{var}}$ to convert the textual descriptions of these variables into a natural language query. We then use this

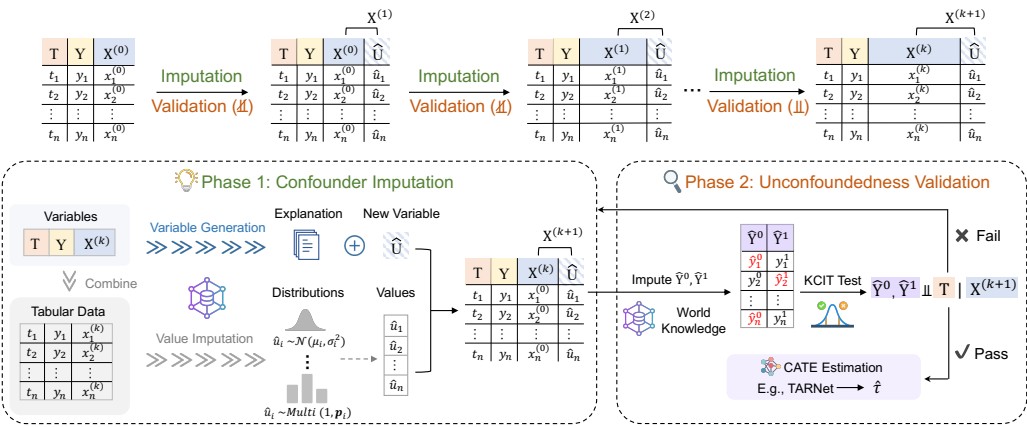

Figure 2: Overview of the ProCI framework. ProCI alternates between two phases: (1) **Confounder Imputation**, where the LLM generates a new confounder and imputes its instance-level values based on current variables. (2) **Unconfoundedness Validation**, where counterfactual outcomes are imputed and a KCIT is applied to assess whether unconfoundedness holds. The process repeats until the test passes, after which any standard estimator can be used for final CATE estimation.

prompt to query the LLM for a candidate confounder variable $\hat{U}$ that is semantically associated with both $T$ and $Y$, while being orthogonal to $X$:

$$\hat{U}, \hat{U}_{\text{exp}} = \text{LLM}(\mathcal{P}_{\text{var}}(X, T, Y)), \tag{3}$$

where $\hat{U}$ is the name or description of the generated confounder, and $\hat{U}_{\text{exp}}$ is the accompanying natural language explanation provided by the LLM. This explanation enhances the interpretability of the generation process and can serve as supporting evidence for the semantic plausibility of $\hat{U}$.

**Value Imputation.** After generating the confounder variable $\hat{U}$, we next impute its values for each individual to obtain an augmented dataset $\hat{\mathcal{D}}_{\text{obs}} = \{x_i, t_i, y_i, \hat{u}_i\}_{i=1}^N$. Direct value imputation, i.e., prompting the LLM to generate $\hat{u}_i$ for each individual via a single-pass query, tends to produce collapsed outputs where many individuals receive similar or identical values. To address this issue, we decompose value imputation into distribution identification and parameter inference as below.

*Distribution Identification.* Intuitively, by leveraging its embedded commonsense and world knowledge, the LLM can reliably identify the distribution type of a variable, for example, inferring that height follows a normal distribution, while birth month is approximately uniform in a community. Therefore, for the generated variable $\hat{U}$, we can confidently ask the LLM to identify a reasonable distribution type $\mathcal{F} \in \mathbb{F} = \{\mathcal{F}_z\}_{z=1}^Z$:

$$\mathcal{F} \sim \text{LLM}(\mathcal{P}_{\text{dist}}(X, T, Y, \hat{U})) \mid \mathcal{F} \in \mathbb{F}. \tag{4}$$

In practice, the candidate distribution types $\mathbb{F}$ include, but are not limited to, Gaussian, Bernoulli, and categorical distributions.

*Parameter Inference.* Once the distribution family $\mathcal{F}$ is selected, we request the LLM to infer individual-specific distribution parameters $\theta_i$ based on each individual's observed features. For example, $\theta_i = (\mu_i, \sigma_i)$ if $\mathcal{F}$ is a normal distribution. We then sample the imputed value $\hat{u}_i$ from the corresponding personalized distribution:

$$\theta_i = \text{LLM}(\mathcal{P}_{\text{param}}(x_i, t_i, y_i)), \quad \hat{u}_i \sim \mathcal{F}(\theta_i). \tag{5}$$

### 3.4 IMPUTATION-BASED UNCONFOUNDEDNESS VALIDATION

While LLMs can help to impute the confounder, it is still challenging to test the validity of the generated confounder. Intuitively, a crucial criterion is the unconfoundedness assumption (Assumption 1). However, in observational data, we can only observe one of two potential outcomes for each individual, making empirical testing of the unconfoundedness assumption seemingly infeasible.

To bridge this gap, we use LLMs to empirically impute the missing outcomes $Y^0$ and $Y^1$ as follows:

$$\hat{y}_i^{1-t_i} = \text{LLM}(\mathcal{P}_{\text{out}}(x_i, u_i, t_i, y_i)). \tag{6}$$

Then, we construct the variables $\hat{Y}^0$ and $\hat{Y}^1$, where for each individual $i$:

$$\hat{y}_i^0 = (1 - t_i) \cdot y_i + t_i \cdot \hat{y}_i^{1-t_i}, \quad \hat{y}_i^1 = t_i \cdot y_i + (1 - t_i) \cdot \hat{y}_i^{1-t_i}. \tag{7}$$

**Rationality Analysis.** Since LLMs embed extensive world knowledge that potentially covers all relevant hidden confounders, they can be viewed as approximately unbiased counterfactual estimators under the unconfoundedness assumption. That is, when conditioned on the full context accessible to the LLM—including both structured and unstructured information—the potential outcomes become independent of the treatment. Formally, we posit that the LLM implicitly conditions on a latent variable $U^*$ such that $(Y^0, Y^1) \perp\!\!\!\perp T \mid X, U^*$, where $U^*$ denotes a sufficient set of hidden confounders encoded in the LLM's prior knowledge.

After obtaining the imputed outcomes $\hat{Y}^0$ and $\hat{Y}^1$, we can statistically test the unconfoundedness assumption using a non-parametric independence test. Specifically, we adopt the Kernel-based Conditional Independence Test (KCIT), which is well-suited for high-dimensional conditioning sets (Pogodin et al., 2024). KCIT evaluates whether the treatment $T$ is conditionally independent of the imputed outcomes $(\hat{Y}^0, \hat{Y}^1)$ given the observed and generated confounders $X$ and $\hat{U}$:

$$\mathbb{I}\left(\texttt{KCIT}\big((\hat{Y}^0, \hat{Y}^1), T \mid X, \hat{U}\big) > \alpha\right) = 1, \tag{8}$$

where $\texttt{KCIT}(\cdot)$ returns a $p$-value, and $\alpha$ is a pre-defined significance level. The indicator function $\mathbb{I}(\cdot)$ outputs 1 if the null hypothesis of conditional independence is not rejected, and 0 otherwise.

Based on the following theorem, we establishe the asymptotic validity of using imputed counterfactual outcomes and LLM-generated confounders in conditional independence testing.

**Theorem 1.** *Under standard regularity conditions on the kernel function and the class of imputation distributions, the KCIT applied to imputed variables satisfies:*

$$\textit{KCIT}\big((\hat{Y}^0, \hat{Y}^1), T \mid X, \hat{U}\big) = \textit{KCIT}\big((Y^0, Y^1), T \mid X, U\big) + o_p(1),$$

*where $o_p(1)$ denotes a term that converges to zero in probability as the sample size increases. Please kindly refer to Appendix C for a detailed proof of this theorem.*

### 3.5 PROGRESSIVE CONFOUNDER IMPUTATION PROCEDURE

Up to now, we have discussed imputing a single confounder and testing its validity. However, in real-world scenarios, hidden confounding often arises from multiple factors, such as demographic and socioeconomic in healthcare (Prosperi et al., 2020; Grootendorst, 2007), many of which are difficult to observe. A naive approach of generating multiple confounders simultaneously may lead to redundancy or semantic overlap. To address this, we propose the complete ProCI framework, which incrementally generates confounders step by step, ensuring both diversity and sufficiency.

Formally, let $X^{(0)} = X$ denote the initial observed covariates. At each iteration $k$, we query the LLM using the current context $(X^{(k-1)}, T, Y)$ to produce a new confounder $\hat{U}$ and update $X$ as:

$$\hat{U}, \hat{U}_{\text{exp}} = \text{LLM}(\mathcal{P}_{\text{var}}(X^{(k-1)}, T, Y)), \quad X^{(k)} = [X^{(k-1)}, \hat{U}]. \tag{9}$$

After each generation step, we apply the empirical unconfoundedness test described in Eq. (14) to assess whether the updated set $X^{(k)}$ sufficiently captures all relevant confounding information. The process terminates at the smallest iteration $k$ such that $X^{(k)}$ passes the test:

$$\min\left\{k \,\middle|\, \texttt{KCIT}\big((\hat{Y}^0, \hat{Y}^1), T \mid X^{(k)}\big) > \alpha\right\}. \tag{10}$$

## 4 EXPERIMENTS

In this section, we conduct experiments to evaluate the effectiveness of the ProCI framework, guided by the following research questions: **[RQ1]** Can imputed confounders from different LLMs improve treatment effect estimation? **[RQ2]** Do the imputed confounders capture additional confounding information beyond observed covariates? **[RQ3]** Is ProCI robust to varying levels of hidden confounding? **[RQ4]** What is the contribution of each ProCI component?

Table 1: Overall comparison of treatment effect estimation performance on the Jobs and Twins datasets. For each base model, the best results across methods are highlighted.

| Datasets | Jobs | | | | Twins | | | |
|---|---|---|---|---|---|---|---|---|
| Test Types | In-sample | | Out-sample | | In-sample | | Out-sample | |
| Methods | $\epsilon_{ATT}$ | $\mathcal{R}_{pol}$ | $\epsilon_{ATT}$ | $\mathcal{R}_{pol}$ | $\epsilon_{ATE}$ | $\epsilon_{PEHE}$ | $\epsilon_{ATE}$ | $\epsilon_{PEHE}$ |
| **S-Learner** | $0.0491_{\pm0.0011}$ | $0.2289_{\pm0.0008}$ | $0.0876_{\pm0.0018}$ | $0.1678_{\pm0.0002}$ | $0.0131_{\pm0.0020}$ | $0.2527_{\pm0.0012}$ | $0.0037_{\pm0.0024}$ | $0.2819_{\pm0.0001}$ |
| +ProCI-4o | $0.0793_{\pm0.0043}$ | $0.2288_{\pm0.0001}$ | $0.0833_{\pm0.0001}$ | $0.1667_{\pm0.0014}$ | $0.0057_{\pm0.0000}$ | $0.2524_{\pm0.0001}$ | $0.0077_{\pm0.0000}$ | $0.2867_{\pm0.0003}$ |
| +ProCI-R1 | $0.0216_{\pm0.0002}$ | $0.2324_{\pm0.0000}$ | $0.0712_{\pm0.0001}$ | $0.1745_{\pm0.0002}$ | $0.0066_{\pm0.0008}$ | $0.2596_{\pm0.0001}$ | $0.0028_{\pm0.0009}$ | $0.2817_{\pm0.0004}$ |
| **PSM** | $0.6197_{\pm0.0000}$ | $0.2707_{\pm0.0000}$ | $0.1259_{\pm0.0015}$ | $0.2192_{\pm0.0013}$ | $0.0457_{\pm0.0006}$ | $0.3399_{\pm0.0007}$ | $0.0840_{\pm0.0000}$ | $0.4027_{\pm0.0001}$ |
| +ProCI-4o | $0.6149_{\pm0.0001}$ | $0.2691_{\pm0.0017}$ | $0.1125_{\pm0.0032}$ | $0.2219_{\pm0.0002}$ | $0.0454_{\pm0.0051}$ | $0.3396_{\pm0.0004}$ | $0.0825_{\pm0.0021}$ | $0.4002_{\pm0.0011}$ |
| +ProCI-R1 | $0.6245_{\pm0.0000}$ | $0.2633_{\pm0.0000}$ | $0.1292_{\pm0.0041}$ | $0.2163_{\pm0.0131}$ | $0.0454_{\pm0.0040}$ | $0.3399_{\pm0.0027}$ | $0.0849_{\pm0.0009}$ | $0.4033_{\pm0.0125}$ |
| **TARNet** | $0.0191_{\pm0.0002}$ | $0.2177_{\pm0.0001}$ | $0.1466_{\pm0.0026}$ | $0.2201_{\pm0.0002}$ | $0.0233_{\pm0.0044}$ | $0.2917_{\pm0.0001}$ | $0.0310_{\pm0.0005}$ | $0.3237_{\pm0.0001}$ |
| +ProCI-4o | $0.0424_{\pm0.0023}$ | $0.2167_{\pm0.0022}$ | $0.3223_{\pm0.0098}$ | $0.2150_{\pm0.0202}$ | $0.0144_{\pm0.0218}$ | $0.2729_{\pm0.0017}$ | $0.0185_{\pm0.0001}$ | $0.3107_{\pm0.0206}$ |
| +ProCI-R1 | $0.0131_{\pm0.0001}$ | $0.2266_{\pm0.0000}$ | $0.0656_{\pm0.0024}$ | $0.2111_{\pm0.0004}$ | $0.0159_{\pm0.0001}$ | $0.2844_{\pm0.0009}$ | $0.0243_{\pm0.0001}$ | $0.3172_{\pm0.0001}$ |
| **CFR-Wass** | $0.0355_{\pm0.0006}$ | $0.2150_{\pm0.0001}$ | $0.1487_{\pm0.0028}$ | $0.2191_{\pm0.0004}$ | $0.0189_{\pm0.0000}$ | $0.2818_{\pm0.0000}$ | $0.0186_{\pm0.0002}$ | $0.3138_{\pm0.0000}$ |
| +ProCI-4o | $0.0300_{\pm0.0009}$ | $0.2085_{\pm0.0001}$ | $0.0402_{\pm0.0003}$ | $0.2151_{\pm0.0005}$ | $0.0094_{\pm0.0001}$ | $0.2729_{\pm0.0001}$ | $0.0178_{\pm0.0001}$ | $0.3110_{\pm0.0001}$ |
| +ProCI-R1 | $0.0303_{\pm0.0010}$ | $0.2264_{\pm0.0042}$ | $0.1141_{\pm0.0067}$ | $0.2088_{\pm0.0004}$ | $0.0215_{\pm0.0002}$ | $0.2796_{\pm0.0061}$ | $0.0282_{\pm0.0003}$ | $0.3119_{\pm0.0032}$ |
| **ESCFR** | $0.0543_{\pm0.0012}$ | $0.2184_{\pm0.0001}$ | $0.2245_{\pm0.0390}$ | $0.2274_{\pm0.0002}$ | $0.0199_{\pm0.0001}$ | $0.2715_{\pm0.0001}$ | $0.0207_{\pm0.0003}$ | $0.3059_{\pm0.0007}$ |
| +ProCI-4o | $0.0369_{\pm0.0004}$ | $0.2174_{\pm0.0001}$ | $0.3225_{\pm0.0622}$ | $0.1891_{\pm0.0002}$ | $0.0191_{\pm0.0001}$ | $0.2700_{\pm0.0032}$ | $0.0297_{\pm0.0029}$ | $0.3045_{\pm0.0001}$ |
| +ProCI-R1 | $0.0130_{\pm0.0001}$ | $0.2219_{\pm0.0000}$ | $0.1294_{\pm0.0072}$ | $0.2101_{\pm0.0008}$ | $0.0087_{\pm0.0001}$ | $0.2684_{\pm0.0002}$ | $0.0132_{\pm0.0001}$ | $0.3038_{\pm0.0000}$ |

## 4.1 EXPERIMENTAL SETUP

**Datasets.** We evaluate on two commonly-used causal benchmarks: **Twins** (Almond et al., 2005) and **Jobs** (LaLonde, 1986). **Twins** contains 8,244 twin pairs from U.S. birth records, with treatment indicating the heavier twin and outcome measuring one-year mortality. The dataset includes 50 demographic and birth-related covariates. Selection bias is introduced following (Louizos et al., 2017). **Jobs** studies the effect of a job training program on employment status. It includes 297 treated individuals, 425 randomized controls, and 2,490 observational controls, with 7 covariates describing demographic and economic features.

**Base Models.** ProCI augments observed data with confounders and thus is model-agnostic. We evaluate it with: (i) meta-learners (**S-Learner** (Künzel et al., 2019)); (ii) matching-based methods (**PSM** (Caliendo & Kopeinig, 2008)); and (iii) representation learning models (**TARNet**, **CFR-Wass** (Shalit et al., 2017), and **ESCFR** (Wang et al., 2023)). For confounder generation, we use two advanced LLMs: **GPT-4o** (Hurst et al., 2024) and **DeepSeek-R1** (Guo et al., 2025).

**Training & Evaluation Protocols.** We use grid search for hyperparameter tuning on validation sets, and adopt consistent settings across shared base models. LLM temperature is fixed at 0.7. All models are implemented in PyTorch 1.10 and trained with Adam optimizer. For both datasets, we split the data into training, validation, and test sets (63:27:10 for Twins, 56:24:20 for Jobs). Twins is evaluated using $\epsilon_{PEHE}$ and $\epsilon_{ATE}$ (Hill, 2011), while Jobs uses $\epsilon_{ATT}$ and policy risk ($\mathcal{R}_{pol}$) (Shalit et al., 2017) for evaluation. Please kindly refer to Appendix D for more experimental details.

## 4.2 [RQ1] CATE PERFORMANCE WITH IMPUTED CONFOUNDERS

To answer RQ1, we apply our proposed ProCI framework using two LLMs, GPT-4o and DeepSeek-R1, denoted as ProCI-4o and ProCI-R1 respectively. We evaluate both variants across a diverse set of base CATE estimators. Each experiment is repeated five times to ensure stability.

**Results.** As shown in Table 1, we can see: $i$) ProCI-augmented estimators consistently outperform base models in both in-sample and out-of-sample settings, demonstrating the effectiveness of our imputed confounders. $ii$) All base model categories benefit from ProCI, with representation-based methods gaining the most, likely due to improved latent variable balancing. $iii$) ProCI-R1 generally outperforms ProCI-4o, indicating that DeepSeek-R1's stronger reasoning leads to more accurate confounder generation and better bias correction. The results from two open-source models, **LLaMA 3-8B** and **Qwen2.5-7B**, are provided in Appendix E.1. Additionally, since the Twins dataset includes ground-truth potential outcomes, we further evaluate the estimation accuracy of the potential outcomes using our ProCI framework in Appendix E.3.

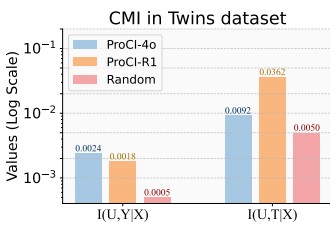

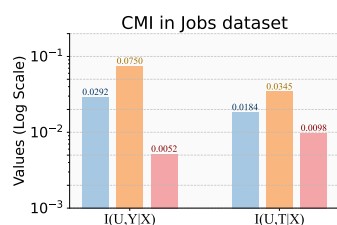

**Confounder 1:** Access to Childcare

**Explanation:** Individuals without reliable childcare may struggle to attend training projects regularly or maintain full-time employment, reducing their chances of program enrollment (treatment) and job placement (outcome).

**Confounder 2:** Criminal Record

**Explanation:** Individuals with a criminal record may be less likely to participate in job training programs due to eligibility restrictions (treatment). Simultaneously, a criminal record can reduce employment opportunities due to employer discrimination (outcome). This confounder is distinct from existing confounders like race or education.

**Confounder 3:** Employment History

**Explanation:** Individuals with strong work histories are more likely to join training to improve skills while those with weaker histories may perceive limited job prospects after training (treatment), affecting their chances of securing employment (outcome).

Figure 3: **(Left)** CMI measures dependence between LLM-generated confounders and treatment/outcome, compared to random confounders. **(Right)** Confounders generated in Jobs demonstrate their semantic relevance and impact on treatment/outcome.

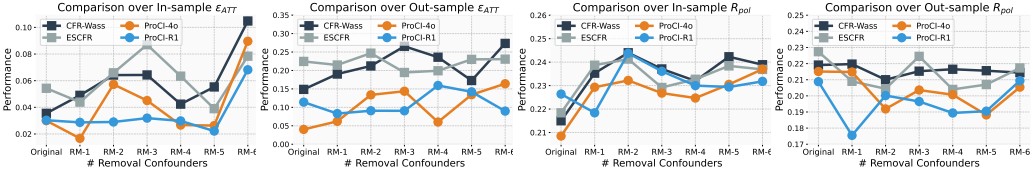

Figure 4: Robustness of ProCI in CATE estimation with varying hidden confounder removal.

### 4.3 [RQ2] CONFOUNDING INFORMATION IN IMPUTED CONFOUNDERS

To answer RQ2, we evaluate whether the confounders imputed by ProCI contain additional information beyond observed covariates. We use conditional mutual information (CMI) to measure the dependency of imputed confounders $U$ with treatment $T$ and outcome $Y$ given observed variables $X$, i.e., $I(U, T|X)$ and $I(U, Y|X)$. Higher CMI values indicate more confounding information.

**Results.** Figure 3 (left) compares imputed confounders from ProCI variants using GPT-4o and DeepSeek-R1 against random confounders generation baseline. The notably higher CMI values confirm that ProCI's confounders carry significant additional information about treatment and outcome beyond observed covariates. In addition, Figure 3 (right) shows three example confounders generated by DeepSeek-R1 on Jobs. These confounders have plausible links to treatment and outcomes and are distinct from observed covariates, highlighting ProCI's ability to uncover meaningful latent confounders omitted by traditional data. We also investigate the impact of LLM's temperature on confounder generation quality in Appendix E.2.

### 4.4 [RQ3] ROBUSTNESS OF PROCI TO HIDDEN CONFOUNDING

In this section, we test the robustness of CATE estimators under hidden confounders. Since hidden confounders are unobservable, we simulate their effects by progressively removing $\{0, 1, 2, 3, 4, 5, 6\}$ confounders from the dataset. This allows us to mimic the impact of latent confounders in a controlled manner. We experiment on Jobs and select CFR-Wass as base model.

**Results.** As shown in Figure 4, when the number of removed confounders increases, the performance of vanilla CATE estimators, such as CFR-Wass and ESCFR, degrades significantly across all metrics. This is expected, since these models are vulnerable to hidden confounder bias, leading to distorted CATE estimations. In contrast, our proposed ProCI framework maintains stable CATE estimation performance despite the progressive removal of confounders from the observed dataset. This robustness is primarily attributed to ProCI's progressive confounder generation capability, which introduces novel and informative confounders to effectively counteract the sparsity induced by the

removal of observed confounders. Furthermore, we also conduct experiments in Appendix E.4 to reveal the matching degree between the generated confounders and the observed covariates.

### 4.5 [RQ4] ABLATION STUDIES OF PROCI COMPONENTS

In this experiment, we conduct ablation studies to evaluate ProCI components. In specific, we tailor three variants: $i$) ProCI w/o DR, which removes distributional reasoning; $ii$) ProCI w/o PI, which eliminates the progressive manner in generating confounders; and $iii$) ProCI w/o UT, which omits the unconfoundedness test. Experiments are conducted on Jobs using GPT-4o as the LLM backbone.

Table 2: Results of ablation studies.

| Methods | In-sample | | Out-sample | |
|---|---|---|---|---|
| | $\epsilon_{ATT}$ | $\mathcal{R}_{pol}$ | $\epsilon_{ATT}$ | $\mathcal{R}_{pol}$ |
| w/o DR | 0.0327 | 0.2178 | 0.1599 | 0.2229 |
| w/o PI | 0.0308 | 0.2223 | 0.1020 | 0.2374 |
| w/o UT | 0.0315 | 0.2218 | 0.0890 | 0.2270 |
| ProCI | **0.0300** | **0.2085** | **0.0402** | **0.2151** |

**Results.** Table 2 shows that removing any component from ProCI reduces performance. Specifically, removing distributional reasoning (w/o DR) leads to less robust estimations due to the corrupted values often generated by LLMs. Removing the progressive imputation process (w/o PI), where all confounders are generated at once, would inevitably introduces semantic overlap. Finally, omitting the unconfoundedness test (w/o UT) can include irrelevant confounders, further degrading the performance of ProCI.

## 5 RELATED WORKS

**Treatment Effect Estimation with Hidden Confounding.** To mitigate hidden confounding, existing approaches generally fall into three categories. Sensitivity analysis methods (Rosenbaum & Rubin, 1983; Robins et al., 2000) assess the potential effect of hidden confounders by deriving bounds on treatment effect estimates, though they rely on fixed, unverifiable assumptions (Franks et al., 2018; Veitch & Zaveri, 2020). Auxiliary variable techniques, including instrumental variables and front-door adjustments (Li et al., 2023; Fulcher et al., 2017; Shah et al., 2023), exploit external or intermediate variables to achieve unbiased estimates, but these depend on unverifiable structural assumptions (Imbens, 2014; Wu et al., 2022; Bellemare et al., 2024). Another line of work resorts to RCT data (Kallus et al., 2018; Hatt et al., 2022; Wu & Yang, 2022) to correct for hidden biases, but their applicability is often constrained by the high costs of RCT data.

**LLMs for treatment effect estimation**. LLMs have recently been explored for causal inference tasks (Liu et al., 2024; Zhao et al., 2024; Chen et al., 2024a; Lin et al., 2024; Wei et al., 2022; Tan, 2023; Chen et al., 2024b; Lin et al., 2023; Vashishtha et al., 2025), especially in estimating treatment effects through prompt-based reasoning (Abdali et al., 2023; Jin et al., 2023; Pawlowski et al., 2023; Dhawan et al., 2024; Imai & Nakamura, 2024). These methods typically focus on extracting causal structures from text or guiding LLMs to simulate interventional queries via carefully crafted prompts. For example, recent works use self-consistency (Abdali et al., 2023), tool augmentation (Pawlowski et al., 2023), and chain-of-thought prompting Jin et al. (2023) to improve causal variable identification and treatment effect estimation. Dhawan et al. (2024) further automates treatment effect estimation by combining LLM outputs with traditional treatment effect estimators.

## 6 CONCLUSION

In this work, we make the first attempt to leverage LLMs to mitigate hidden confounding in treatment effect estimation. We propose ProCI, a framework that progressively elicits LLMs to generate, impute, and validate hidden confounders using both structured and unstructured information. By incorporating distribution-aware imputation and an empirical unconfoundedness test grounded in the world knowledge embedded in LLMs, ProCI provides a robust and scalable solution for treatment effect estimation under hidden confounding. Extensive experiments across diverse datasets and LLMs demonstrate the effectiveness of our approach. Our results show that ProCI significantly improves treatment effect estimation, offering a novel solution to the challenge of hidden confounding. This work offers a new perspective on using LLMs as knowledge-rich tools for treatment effect estimation under hidden confounding.

ETHICS STATEMENT

We affirm that this research fully complies with the ICLR Code of Ethics. The study does not involve human subjects, and all datasets utilized are publicly available, containing no sensitive or personal information. We have carefully considered the ethical implications of our work, particularly regarding fairness and the potential risks associated with the misuse of LLMs. Our commitment remains focused on ensuring that this research makes a positive and responsible contribution to both the academic community and society.

REPRODUCIBILITY STATEMENT

To ensure reproducibility, the code for implementing the ProCI framework will be provided as supplemental material upon submission. Detailed instructions for running the experiments, along with the prompts used in the confounder imputation process, are included in Appendix F. Information regarding the datasets, experimental setup, and data processing steps can be found in Section 4.1 and Appendix D. The theoretical assumptions and proofs are provided in Appendix B and C. These resources collectively ensure that the methods and results presented can be accurately replicated.

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

CONTENTS

## A  ACKNOWLEDGMENT OF LLM USAGE

In the preparation of this manuscript, large language models (LLMs) were solely employed for tasks related to language refinement, such as fixing typographical errors, improving grammatical accuracy, and enhancing sentence structure. While LLMs were employed as a component of the technical framework, they were not central to the core research processes. Specifically, LLMs played no role in the formulation of research ideas, data analysis, or interpreting the findings. The authors bear full responsibility for the scientific content presented.

## B  IN-DEPTH CAUSAL INFERENCE PRELIMINARIES

This section provides additional background on causal inference, which is particularly intended to assist readers who may be less familiar with the foundational concepts or technical aspects of treatment effect estimation from observational data.

We begin by introducing key definitions and assumptions underlying causal inference from observational data. For an individual characterized by covariates $x$, there are two potential outcomes: $Y^1$ if the individual receives treatment and $Y^0$ if assigned to control. The Conditional Average Treatment Effect (CATE) captures the expected difference in outcomes between these two scenarios:

**Definition 1.** *The CATE for individuals with covariates $x$ is defined as*

$$\tau(x) = \mathbb{E}[Y^1 - Y^0 \mid X = x], \tag{11}$$

*where $X$ denotes the covariate variable, and $Y^1$, $Y^0$ are the potential outcomes under treatment and control, respectively.*

Estimating CATE from observational data poses two primary challenges:

1. *Missing counterfactuals*: For each individual, we only observe the outcome corresponding to the assigned treatment. The unobserved counterfactual remains inaccessible.

2. *Selection bias*: Treatment assignment may depend on covariates related to the outcome, leading to systematic differences between treated and control groups.

To address these challenges, Pearl & Mackenzie (2018) proposed a two-stage framework. The first stage, *identification*, aims to express causal quantities, such as $\tau(x)$, in terms of observed data using assumptions and adjustment formulas. Identification is not always guaranteed, and depends on the following assumptions:

**Assumption 2** (Ignorability / Unconfoundedness). *The treatment assignment is independent of the potential outcomes given the covariates, i.e., $Y^1, Y^0 \perp\!\!\!\perp T \mid X = x$.*

**Assumption 3** (Positivity). *For any value $x$, both treatment and control must have non-zero probability, i.e., $0 < P(T = t \mid X = x) < 1$ for all $t \in \{0, 1\}$.*

**Assumption 4** (SUTVA). *The potential outcomes for any individual are unaffected by others' treatment assignments, and each treatment corresponds to a single well-defined outcome.*

**Assumption 5** (Consistency). *The observed outcome equals the potential outcome under the treatment actually received.*

Once identification is established, the second stage, *estimation*, transforms the causal estimand into a statistical estimand that can be computed from the data:

$$\begin{aligned}
\mathbb{E}[Y^1 - Y^0 \mid X = x] &= \mathbb{E}[Y^1 \mid X = x] - \mathbb{E}[Y^0 \mid X = x] \\
&\overset{(1)}{=} \mathbb{E}[Y^1 \mid X = x, T = 1] - \mathbb{E}[Y^0 \mid X = x, T = 0] \\
&\overset{(2)}{=} \mathbb{E}[Y \mid X = x, T = 1] - \mathbb{E}[Y \mid X = x, T = 0],
\end{aligned} \tag{12}$$

where step (1) uses Assumption 2, and step (2) additionally relies on Assumptions 3, 4, and 5.

In practice, numerous approaches have been developed to estimate the last quantity. Classical methods include matching strategies (Caliendo & Kopeinig, 2008), which pair treated and control units

with similar covariates, and meta-learners (Künzel et al., 2019), which adapt supervised learning techniques to causal inference tasks. More recent advances leverage deep learning, notably representation learning methods (Assaad et al., 2021; Wu et al., 2023), which aim to mitigate selection bias by learning balanced latent spaces. A prominent example is Counterfactual Regression (CFR) (Shalit et al., 2017), which introduces regularization terms based on distributional distances—such as the Wasserstein distance or the maximum mean discrepancy—to align representations between treatment groups.

Despite their success, these approaches heavily rely on the unconfoundedness assumption, which is often violated in real-world datasets where hidden confounding factors may influence both treatment and outcome. While some recent methods attempt to account for hidden confounding, they typically depend on strong structural assumptions, additional proxies, or access to experimental (RCT) data—which may be costly or unavailable in practice.

In this paper, we take a new direction by exploring the potential of large language models (LLMs) to assist in imputing latent confounders from observational text and tabular data. Specifically, we propose a framework ProCI that leverages LLMs' implicit knowledge and generative ability to infer proxy variables that encode hidden causal information—thus helping to relax the unconfoundedness assumption and improve the robustness of causal estimates in the presence of hidden confounding.

## C    PROOF OF THEOREM 1

In Theorem 1, we aim to employ the kernel-based conditional independence test (KCIT) to assess whether the potential outcomes are conditionally independent of the treatment variable, given both observed and latent covariates. Specifically, we test whether:

$$\boldsymbol{Y} = (Y^0, Y^1) \perp\!\!\!\perp T \mid X, U$$

**Hypotheses.**

- **Null Hypothesis** ($H_0$): $\boldsymbol{Y} \perp\!\!\!\perp T \mid X, U$ — the potential outcomes are conditionally independent of treatment given covariates.
- **Alternative Hypothesis** ($H_1$): $\boldsymbol{Y} \not\perp\!\!\!\perp T \mid X, U$ — there exists residual dependence between treatment and outcomes after conditioning on covariates.

**Test Statistic.**    KCIT estimates the squared Hilbert-Schmidt norm of the partial cross-covariance operator between $\boldsymbol{Y} = (Y^0, Y^1)$ and $T$ given $(X, U)$. Let $n$ be the sample size. For two random vectors $X, Y \in \mathcal{X} \times \mathcal{Y}$, define the cross-variance operator as $\langle g, \Sigma_{YX} f \rangle = \mathbb{E}_{XY}(f(X)g(Y)) - \mathbb{E}_X f(X)\mathbb{E}_Y g(Y)$ for $f \in \mathcal{H}_X$ and $g \in \mathcal{H}_Y$, the RKHS of $\mathcal{X}$ and $\mathcal{Y}$ respectively. The cross-variance operator is estimated via $\hat{\Sigma}_{YX} = \frac{1}{n} Tr(\tilde{K}_X \tilde{K}_Y)$, with $\tilde{K}_X = H K_X H$, $H = I - \frac{1}{n}\mathbf{1}\mathbf{1}^\top$ and the (i,j)-th entry of $K_X$ is $k(x_i, x_j)$. The KCIT operator is estimated via:

$$\hat{\Sigma}_{\boldsymbol{Y}T|(X,U)} = \hat{\Sigma}_{\boldsymbol{Y}T} - \hat{\Sigma}_{\boldsymbol{Y}Z}(\hat{\Sigma}_{ZZ} + \gamma I)^{-1}\hat{\Sigma}_{ZT} \tag{13}$$

with $Z = (X, U)$, and $\gamma > 0$ is a regularization parameter. Finally, the KCIT test statistics is constructed by

$$\text{KCIT}(\boldsymbol{Y}, T \mid (X, U)) = \frac{1}{n} Tr(\hat{\Sigma}_{\boldsymbol{Y}T|(X,U)}). \tag{14}$$

**Lemma 1** (Theorem 1 in (Amini & Razaee, 2021)). *Let $X_i \in \text{LC}(\mu_i, \Sigma_i, \omega), i = 1, \ldots, n$, be a collection of independent random vectors from the LC distribution class defined in Amini & Razaee (2021), and let $K = K(X)$ be the kernel matrix for an L-Lipschitz kernel function $k(x_1, x_2)$, i.e. $|k(x_1, x_2) - k(y_1, y_2)| \leq L(\|x_1 - y_1\| + \|x_2 - y_2\|)$. Then, for some universal constant $c > 0$, with probability at least $1 - \exp(-ct^2)$,*

$$\|K - \mathbb{E}K\| \leq 2L\omega\sigma_\infty(Cn + \sqrt{n}t),$$

*where $\sigma_\infty^2 := \max_i \|\Sigma_i\|$ and $C = c^{-1/2}$.*

**Lemma 2.** *Let $\hat{Y}_i, \hat{Z}_i$ be samples from LC classes $\text{LC}(\mu_{\boldsymbol{Y}}, \Sigma_{Yi}, \omega)$ and $\text{LC}(\mu_Z, \Sigma_{Zi}, \omega)$ respectively[2], such that $\|\mathbb{E}\tilde{K}_{\hat{\boldsymbol{Y}}} - \tilde{K}_{\boldsymbol{Y}}\| = o_p(1)$ and $\|\mathbb{E}\tilde{K}_{\hat{Z}} - \tilde{K}_Z\| = o_p(1)$. Then, for Gaussian RBF*

---

[2]Here $\hat{Z} = (X, \hat{U})$, where $\hat{U}$ is the estimation of latent confounders.

*such that $\sigma \to \infty$ as $n \to \infty$, we have*

$$\|\hat{\Sigma}_{\hat{\boldsymbol{Y}}T} - \hat{\Sigma}_{\boldsymbol{Y}T}\| = o_p(1)$$
$$\|\hat{\Sigma}_{\hat{\boldsymbol{Y}}\hat{Z}} - \hat{\Sigma}_{\boldsymbol{Y}Z}\| = o_p(1)$$
$$\|\hat{\Sigma}_{\hat{Z}T} - \hat{\Sigma}_{ZT}\| = o_p(1) \qquad (15)$$
$$\|\hat{\Sigma}_{\hat{Z}\hat{Z}} - \hat{\Sigma}_{ZZ}\| = o_p(1)$$

*with the last equation implies $\|\hat{\Sigma}_{\hat{Z}\hat{Z}}^{-1} - \hat{\Sigma}_{ZZ}^{-1}\| = o_p(1)$.*

*Proof of Lemma 2.* From Lemma 1, with probability at least $1 - \exp(-ct^2)$,
$$\|K_{\hat{\boldsymbol{Y}}} - \mathbb{E}K_{\hat{\boldsymbol{Y}}}\| \leq 2L\omega\sigma_\infty(Cn + \sqrt{n}t).$$

Therefore, since $L = o(\sigma^{-1}) = o(1)$ as $n$ tends to infinity, $\|\tilde{K}_{\hat{\boldsymbol{Y}}} - \mathbb{E}\tilde{K}_{\hat{\boldsymbol{Y}}}\| \leq \|H\|^2 \|K_{\hat{\boldsymbol{Y}}} - \mathbb{E}K_{\hat{\boldsymbol{Y}}}\| = o_p(n)$. From above we have

$$\|\hat{\Sigma}_{\hat{\boldsymbol{Y}}T} - \hat{\Sigma}_{\boldsymbol{Y}T}\| = \frac{1}{n}Tr(\tilde{K}_X(\tilde{K}_{\hat{\boldsymbol{Y}}} - \tilde{K}_{\boldsymbol{Y}}))$$
$$\leq \frac{1}{n}\|\tilde{K}_X\| \cdot \|\tilde{K}_{\hat{\boldsymbol{Y}}} - \tilde{K}_{\boldsymbol{Y}}\|$$
$$\leq \frac{1}{n}\|\tilde{K}_X\| \cdot \left(\|\tilde{K}_{\hat{\boldsymbol{Y}}} - \mathbb{E}\tilde{K}_{\hat{\boldsymbol{Y}}}\| + \|\mathbb{E}\tilde{K}_{\hat{\boldsymbol{Y}}} - \tilde{K}_{\boldsymbol{Y}}\|\right)$$
$$= o_p(1),$$

which proves the first equation in Eq. (15). The second equation comes from the fact that

$$\|\hat{\Sigma}_{\hat{\boldsymbol{Y}}\hat{Z}} - \hat{\Sigma}_{\boldsymbol{Y}Z}\| = \frac{1}{n}Tr(\tilde{K}_{\hat{\boldsymbol{Y}}}(\tilde{K}_{\hat{Z}} - \tilde{K}_Z) + \tilde{K}_Z(\tilde{K}_{\hat{\boldsymbol{Y}}} - \tilde{K}_{\boldsymbol{Y}}))$$
$$\leq \frac{1}{n}\left(\|\tilde{K}_{\hat{\boldsymbol{Y}}}\| \cdot \|\tilde{K}_{\hat{Z}} - \tilde{K}_Z\| + \|\tilde{K}_Z\| \cdot \|\tilde{K}_{\hat{\boldsymbol{Y}}} - \tilde{K}_{\boldsymbol{Y}}\|\right)$$
$$= o_p(1).$$

with the last equation resulting from $\|\tilde{K}_{\hat{\boldsymbol{Y}}} - \tilde{K}_{\boldsymbol{Y}}\| = o_p(n)$ and $\|\tilde{K}_{\hat{Z}} - \tilde{K}_Z\| = o_p(n)$. The third equation in Eq. (15) comes from the same deduction as for the first equation, and the last equation comes from the same deduction for the second equation.

Finally, the conclusion on the inverse matrix is straightforward observing that

$$(\hat{\Sigma}_{\hat{Z}\hat{Z}} + \mathcal{E})^{-1} = \hat{\Sigma}_{\hat{Z}\hat{Z}}^{-1} - \hat{\Sigma}_{\hat{Z}\hat{Z}}^{-1}\mathcal{E}\hat{\Sigma}_{\hat{Z}\hat{Z}}^{-1} + O_p(\|\mathcal{E}\|) = \hat{\Sigma}_{ZZ}^{-1} + o_p(1)$$

with $\|\mathcal{E}\| = o_p(1)$. $\qquad \square$

**Theorem 2.** *Under standard regularity conditions on the kernel function and the class of imputation distributions, the KCIT applied to imputed variables satisfies:*
$$\text{KCIT}\big((\hat{Y}^0, \hat{Y}^1), T \mid X, \hat{U}\big) = \text{KCIT}\big((Y^0, Y^1), T \mid X, U\big) + o_p(1),$$
*where $o_p(1)$ denotes a term that converges to zero in probability as the sample size increases.*

**Proof of Theorem 1.** Based on Lemma 2 and Eq. (13), we have

$$\hat{\Sigma}_{\hat{\boldsymbol{Y}}T|\hat{Z}} = \hat{\Sigma}_{\hat{\boldsymbol{Y}}T} - \hat{\Sigma}_{\hat{\boldsymbol{Y}}\hat{Z}}(\hat{\Sigma}_{\hat{Z}\hat{Z}} + \gamma I)^{-1}\hat{\Sigma}_{\hat{Z}T}$$
$$= \hat{\Sigma}_{\boldsymbol{Y}T} - \hat{\Sigma}_{\boldsymbol{Y}Z}(\hat{\Sigma}_{ZZ} + \gamma I)^{-1}\hat{\Sigma}_{ZT} + o_p(1)$$
$$= \hat{\Sigma}_{\boldsymbol{Y}T|Z} + o_p(1).$$

Therefore, based on Eq. (14), we have

$$|\text{KCIT}\big((Y^0, Y^1), T \mid X, U\big) - \text{KCIT}\big((\hat{Y}^0, \hat{Y}^1), T \mid X, \hat{U}\big)|$$
$$= \frac{1}{n}|Tr(\hat{\Sigma}_{\hat{\boldsymbol{Y}}T|\hat{Z}} - \hat{\Sigma}_{\boldsymbol{Y}T|Z})|$$
$$\leq \|\hat{\Sigma}_{\hat{\boldsymbol{Y}}T|\hat{Z}} - \hat{\Sigma}_{\boldsymbol{Y}T|Z}\|$$
$$= o_p(1),$$

which proves the theorem.

Table 3: Hyperparameter search space used in all experiments.

| Hyperparameter | Search Range | Description |
|---|---|---|
| lr | $\{10^{-5}, 10^{-4}, 10^{-3}, 10^{-2}, 10^{-1}\}$ | learning rate |
| bs | $\{16, 32, 64, 128\}$ | batch size |
| $\lambda$ | $\{10^{-4}, 10^{-3}, 10^{-2}, 10^{-1}, 1\}$ | loss balancing coefficient |
| $d_\phi$ | $\{16, 32, 64\}$ | hidden dimension in encoder network $\phi$ |
| $d_h$ | $\{16, 32, 64\}$ | hidden dimension in outcome heads $h_0$ and $h_1$ |

## D FURTHER EXPERIMENTAL DETAILS

### D.1 DATASET

**Twins.** The Twins dataset is derived from all recorded twin births in the United States between 1989 and 1991 (Almond et al., 2005). We focus on twin pairs where both individuals weighed less than 2000 grams at birth. Each instance contains 50 pre-treatment covariates related to parental characteristics, pregnancy conditions, and birth outcomes. The treatment assignment is defined such that $T = 1$ corresponds to the heavier twin and $T = 0$ to the lighter one. The outcome variable $Y$ is one-year mortality.

After removing records with missing values, the resulting dataset comprises 8,244 samples. Since data for both twins in each pair is available, we observe outcomes under both treatment assignments ($T = 0$ and $T = 1$). To emulate an observational setting, we simulate unobserved counterfactuals by selectively masking one twin per pair. When this selection is randomized, the data mimics a randomized controlled trial. In specific, we model confounding via a proxy variable, where we assign treatment based on a single feature—GESTAT10—which represents gestational age in 10 categories. Formally, treatment is drawn as: $T_i \mid X_i, Z_i \sim \text{Bern}\left(\sigma\left(W_o^\top X_i + W_h(Z_i/10 - 0.1)\right)\right)$, where $W_o \sim \mathcal{N}(0, 0.1 \cdot I)$ and $W_h \sim \mathcal{N}(5, 0.1)$. Here, $X_i$ denotes the 49 non-GESTAT10 features and $Z_i$ is the GESTAT10 value for unit $i$.

**Jobs.** This dataset combines the Lalonde randomized experiment (297 treated and 425 control units) with an observational sample from the PSID (2,490 control units) (LaLonde, 1986). Each record includes 7 covariates such as age, education level, ethnicity, and prior earnings. The outcome reflects post-intervention employment status. By merging the experimental and observational subsets, we can introduce the selection bias between treated and control groups, making this dataset useful for evaluating robustness to such bias.

### D.2 TRAINING AND EVALUATION PROTOCOLS

**Training Protocol.** All models are optimized using grid search based on validation performance. The learning rate and batch size are tuned over predefined discrete sets: $\{10^{-5}, 10^{-4}, 10^{-3}, 10^{-2}, 10^{-1}\}$ for learning rates and $\{16, 32, 64, 128\}$ for batch sizes. For methods involving balancing losses (CFR-Wass, CFR-MMD, and ESCFR), the regularization weight $\lambda$ is selected from $\{10^{-4}, 10^{-3}, 10^{-2}, 10^{-1}, 1\}$ to control the trade-off between outcome prediction and representation alignment. Table 3 summarizes the full hyperparameter configuration space used during training. All models, including the baselines and our proposed method, are tuned under the same conditions to ensure fair comparison.

Training is performed for a maximum of 200 epochs, with early stopping applied based on validation loss. Specifically, we stop the training process if no improvement is observed within 30 consecutive epochs, which helps prevent overfitting—particularly relevant for datasets like Twins, where ground-truth outcomes are fully known. All experiments are implemented using PyTorch 1.10 and trained with the Adam optimizer. Hardware used includes an NVIDIA A40 GPU and an Intel(R) Xeon(R) Gold 5318Y CPU at 2.10GHz.

**Evaluation Protocol.** For the Twins dataset, where the distributions of potential outcomes are available, we evaluate model performance using two metrics: the Precision in Estimation of Heterogeneous Effect ($\epsilon_{PEHE}$) and the Average Treatment Effect error ($\epsilon_{ATE}$) (Hill, 2011).

The PEHE is defined as:

$$\epsilon_{PEHE} = \frac{1}{N} \sum_{i=1}^{N} \left( \mathbb{E}_{(y_i^0, y_i^1) \sim \mathcal{P}_{\mathbf{Y}|\mathbf{x}_i}} \left( y_i^1 - y_i^0 \right) - \left( \hat{y}_i^1 - \hat{y}_i^0 \right) \right)^2 ,$$

where $\hat{y}_i^0$ and $\hat{y}_i^1$ denote the estimated outcomes under control and treatment, respectively, and $y_i^0$ and $y_i^1$ represent the corresponding true outcomes.

For the ATE error, we compute it as:

$$\epsilon_{ATE} = \left| \frac{1}{N} \sum_{i=1}^{N} \left( y_i^1 - y_i^0 \right) - \frac{1}{N} \sum_{i=1}^{N} \left( \hat{y}_i^1 - \hat{y}_i^0 \right) \right| .$$

Lower values of $\epsilon_{PEHE}$ and $\epsilon_{ATE}$ indicate better estimation performance.

For the Jobs dataset, where ground-truth ITE is not available, we use two metrics: policy risk $\mathcal{R}_{pol}$ (Shalit et al., 2017) and the error in estimating the Average Treatment effect on the Treated ($\epsilon_{ATT}$).

Policy risk is defined as:

$$\mathcal{R}_{pol} = 1 - \left( \mathbb{E}[Y^1 \mid \pi(x) = 1] \cdot \mathbb{P}(\pi(x) = 1) + \mathbb{E}[Y^0 \mid \pi(x) = 0] \cdot \mathbb{P}(\pi(x) = 0) \right) ,$$

where $\pi(x) = 1$ if $\hat{y}^1 - \hat{y}^0 > 0$, and $\pi(x) = 0$ otherwise.

We estimate this metric using only units from the randomized component of the dataset:

$$
\begin{aligned}
\mathcal{R}_{pol} = 1 - \Bigg( & \frac{1}{|A^1 \cap T^1 \cap E|} \sum_{\mathbf{x}_i \in A^1 \cap T^1 \cap E} y_i^1 \cdot \frac{|A^1 \cap E|}{|E|} \\
& + \frac{1}{|A^0 \cap T^0 \cap E|} \sum_{\mathbf{x}_i \in A^0 \cap T^0 \cap E} y_i^0 \cdot \frac{|A^0 \cap E|}{|E|} \Bigg)
\end{aligned}
\tag{16}
$$

with $E$ denoting the randomized experiment set, $A^1 = \{\mathbf{x}_i : \hat{y}_i^1 - \hat{y}_i^0 > 0\}$, $A^0 = \{\mathbf{x}_i : \hat{y}_i^1 - \hat{y}_i^0 < 0\}$, $T^1 = \{\mathbf{x}_i : t_i = 1\}$, and $T^0 = \{\mathbf{x}_i : t_i = 0\}$. A lower value of $\mathcal{R}_{pol}$ indicates that the CATE estimation method provides better support for the decision-making strategy.

We also report $\epsilon_{ATT}$ as:

$$\epsilon_{ATT} = \left| \frac{1}{N_1} \sum_{i:t_i=1} \left( y_i^1 - y_i^0 \right) - \frac{1}{N_1} \sum_{i:t_i=1} \left( \hat{y}_i^1 - \hat{y}_i^0 \right) \right| ,$$

where $N_1$ is the number of treated units in the randomized group. A lower $\epsilon_{\text{ATT}}$ indicates more accurate treatment effect estimation for the treated population.

### D.3 BASE MODELS

Since our proposed ProCI framework is model-agnostic and only augments the original dataset with new confounders, it can be flexibly combined with a variety of existing CATE estimation methods. In our experiments, we treat several well-established and state-of-the-art CATE estimators as base models to assess how their performance improves when equipped with the confounders generated by ProCI.

We consider representative methods from three major categories: meta-learning, matching-based, and representation-based approaches.

*i)* **Meta-learners**: These methods differ in how they handle treatment information. The **S-Learner** (Künzel et al., 2019) uses a single model that includes treatment as a feature.

*ii)* **Matching-based methods**: We include propensity score matching (**PSM**) (Caliendo & Kopeinig, 2008), followed by regression on the matched samples. Propensity scores in PSM are estimated using logistic regression, consistent with the implementation in (Caliendo & Kopeinig, 2008).

Table 4: Overall performance comparison of treatment effect estimation between base models and their enhanced versions with ProCI-La (LLaMA 3-8B) and ProCI-Qw (Qwen2.5-7B). Best-performing results across all methods are highlighted.

| Datasets | Jobs | | | | Twins | | | |
|---|---|---|---|---|---|---|---|---|
| Test Types | In-sample | | Out-sample | | In-sample | | Out-sample | |
| Methods | $\epsilon_{ATT}$ | $\mathcal{R}_{pol}$ | $\epsilon_{ATT}$ | $\mathcal{R}_{pol}$ | $\epsilon_{ATE}$ | $\epsilon_{PEHE}$ | $\epsilon_{ATE}$ | $\epsilon_{PEHE}$ |
| **S-Learner** | $0.0491_{\pm0.0011}$ | $0.2289_{\pm0.0008}$ | $0.0876_{\pm0.0018}$ | $0.1678_{\pm0.0002}$ | $0.0131_{\pm0.0020}$ | $0.2527_{\pm0.0012}$ | $0.0037_{\pm0.0024}$ | $0.2819_{\pm0.0001}$ |
| +ProCI-La | $0.0227_{\pm0.0004}$ | $0.2253_{\pm0.0013}$ | $0.1009_{\pm0.0001}$ | $0.1648_{\pm0.0002}$ | $0.0117_{\pm0.0001}$ | $0.2536_{\pm0.0000}$ | $0.0035_{\pm0.0029}$ | $0.2833_{\pm0.0032}$ |
| +ProCI-Qw | $0.0320_{\pm0.0007}$ | $0.2301_{\pm0.0000}$ | $0.0832_{\pm0.0001}$ | $0.1697_{\pm0.0010}$ | $0.0068_{\pm0.0013}$ | $0.2518_{\pm0.0027}$ | $0.0050_{\pm0.0004}$ | $0.2846_{\pm0.0012}$ |
| **PSM** | $0.6197_{\pm0.0000}$ | $0.2707_{\pm0.0000}$ | $0.1259_{\pm0.0015}$ | $0.2192_{\pm0.0013}$ | $0.0457_{\pm0.0006}$ | $0.3399_{\pm0.0007}$ | $0.0840_{\pm0.0000}$ | $0.4027_{\pm0.0001}$ |
| +ProCI-La | $0.6185_{\pm0.0008}$ | $0.2716_{\pm0.0003}$ | $0.1126_{\pm0.0002}$ | $0.2181_{\pm0.0001}$ | $0.0460_{\pm0.0012}$ | $0.3394_{\pm0.0004}$ | $0.0845_{\pm0.0000}$ | $0.4026_{\pm0.0000}$ |
| +ProCI-Qw | $0.6173_{\pm0.0000}$ | $0.2691_{\pm0.0001}$ | $0.0992_{\pm0.0003}$ | $0.2191_{\pm0.0035}$ | $0.0455_{\pm0.0037}$ | $0.3401_{\pm0.0000}$ | $0.0818_{\pm0.0001}$ | $0.4017_{\pm0.0000}$ |
| **TARNet** | $0.0191_{\pm0.0002}$ | $0.2177_{\pm0.0001}$ | $0.1466_{\pm0.0026}$ | $0.2201_{\pm0.0002}$ | $0.0233_{\pm0.0044}$ | $0.2917_{\pm0.0001}$ | $0.0310_{\pm0.0005}$ | $0.3237_{\pm0.0001}$ |
| +ProCI-La | $0.0163_{\pm0.0021}$ | $0.2059_{\pm0.0027}$ | $0.1179_{\pm0.0032}$ | $0.2139_{\pm0.0011}$ | $0.0190_{\pm0.0021}$ | $0.2755_{\pm0.0013}$ | $0.0270_{\pm0.0019}$ | $0.3043_{\pm0.0012}$ |
| +ProCI-Qw | $0.0129_{\pm0.0001}$ | $0.2005_{\pm0.0000}$ | $0.0761_{\pm0.0010}$ | $0.2201_{\pm0.0032}$ | $0.0101_{\pm0.0000}$ | $0.2775_{\pm0.0014}$ | $0.0092_{\pm0.0001}$ | $0.3123_{\pm0.0002}$ |
| **CFR-Wass** | $0.0355_{\pm0.0006}$ | $0.2150_{\pm0.0001}$ | $0.1487_{\pm0.0028}$ | $0.2191_{\pm0.0004}$ | $0.0189_{\pm0.0000}$ | $0.2818_{\pm0.0000}$ | $0.0186_{\pm0.0002}$ | $0.3138_{\pm0.000}$ |
| +ProCI-La | $0.0312_{\pm0.0003}$ | $0.2023_{\pm0.0001}$ | $0.1373_{\pm0.0073}$ | $0.1986_{\pm0.0004}$ | $0.0132_{\pm0.0001}$ | $0.2762_{\pm0.0001}$ | $0.0150_{\pm0.0001}$ | $0.3073_{\pm0.0001}$ |
| +ProCI-Qw | $0.0295_{\pm0.0004}$ | $0.2029_{\pm0.0001}$ | $0.1389_{\pm0.0068}$ | $0.2140_{\pm0.0002}$ | $0.0137_{\pm0.0000}$ | $0.2724_{\pm0.0000}$ | $0.0110_{\pm0.0001}$ | $0.3041_{\pm0.0002}$ |
| **ESCFR** | $0.0543_{\pm0.0012}$ | $0.2184_{\pm0.0001}$ | $0.2245_{\pm0.0390}$ | $0.2274_{\pm0.0002}$ | $0.0199_{\pm0.0001}$ | $0.2715_{\pm0.0001}$ | $0.0207_{\pm0.0003}$ | $0.3059_{\pm0.0007}$ |
| +ProCI-La | $0.0362_{\pm0.0005}$ | $0.2074_{\pm0.0000}$ | $0.1218_{\pm0.0038}$ | $0.2055_{\pm0.0015}$ | $0.0160_{\pm0.0001}$ | $0.2714_{\pm0.0002}$ | $0.0177_{\pm0.0001}$ | $0.3030_{\pm0.0001}$ |
| +ProCI-Qw | $0.0317_{\pm0.0008}$ | $0.2154_{\pm0.0011}$ | $0.2114_{\pm0.0430}$ | $0.2239_{\pm0.0033}$ | $0.0131_{\pm0.0001}$ | $0.2699_{\pm0.0021}$ | $0.0062_{\pm0.0000}$ | $0.2989_{\pm0.0001}$ |

*iii*) **Representation-based methods**: This includes **TARNet** (Shalit et al., 2017), which employs a shared feature representation with separate heads for predicting potential outcomes; **CFR-Wass** (Shalit et al., 2017), which adds distributional regularization using the Wasserstein metric; and **ESCFR** (Wang et al., 2023), which leverages unbalanced optimal transport to achieve mini-batch-level representation balance and robustness to outliers.

# E  ADDITIONAL EXPERIMENTAL RESULTS

## E.1  EFFECTIVENESS OF PROCI WITH OPEN-SOURCE LANGUAGE MODELS

**Experimental Setup.** To further investigate the generalizability of ProCI under different LLM architectures, we introduce two additional models: LLaMA 3–8B and Qwen2.5–7B, denoted as **ProCI-La** and **ProCI-Qw**, respectively. These models are selected for their competitive reasoning capabilities and open accessibility. ProCI-La is based on Meta's LLaMA 3 series (Grattafiori et al., 2024), a dense decoder-only transformer optimized for instruction-following. ProCI-Qw leverages Alibaba's Qwen2.5 family (Qwen et al., 2025), which has shown strong performance in multi-lingual and causal reasoning tasks.

We apply the same ProCI framework using LLaMA 3–8B and Qwen2.5–7B as the confounder generators. All settings (e.g., temperature = 0.7, prompt structure, distribution identification and imputation, progressive confounder generation) remain consistent with the original experiments to ensure fair comparisons. The resulting augmented datasets are then passed into the same downstream CATE estimators as in the original setup.

**Results.** As shown in Table 4, both ProCI-La and ProCI-Qw significantly improve CATE estimation performance over their corresponding base models, confirming the effectiveness of using LLM-generated hidden confounders even beyond proprietary GPT models. Notably, the improvements are consistent across different types of base estimators, with the following observations:

- **ProCI-La, based on LLaMA 3–8B, achieves strong performance gains on both in-sample and out-of-sample evaluations.** Its improvements are particularly evident when paired with representation-based base models such as TARNet and CFR-Wass, indicating that LLaMA 3's semantic reasoning helps uncover latent variables that enhance feature balancing in the learned representations.

- **ProCI-Qw, leveraging Qwen2.5–7B, also brings stable improvements over base models.** While its performance slightly lags behind ProCI-La in some cases, it still consistently

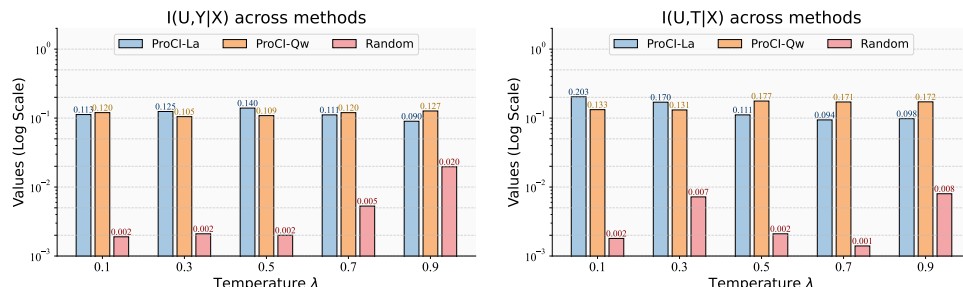

Figure 5: **Conditional mutual information (CMI) values across different temperatures for confounders generated by ProCI-La (LLaMA 3–8B), ProCI-Qw (Qwen2.5–7B), and a Random baseline.** The left plot shows $I(U, Y \mid X)$, and the right plot shows $I(U, T \mid X)$, with all values on a log scale. ProCI methods consistently outperform the Random baseline, with peak informativeness generally occurring at moderate temperatures (0.5–0.7).

 enhances treatment effect estimation, especially under models sensitive to unobserved confounding. This suggests Qwen2.5 can effectively contribute causal priors despite its smaller scale.

- **Across both models, we observe that representation learning-based estimators benefit the most from ProCI augmentation.** These models are designed to learn balanced representations of treated and control groups, and the inclusion of high-quality confounders improves this balancing, thereby reducing estimation bias and variance more effectively.

These findings demonstrate that ProCI is model-agnostic and remains effective when paired with open-source, instruction-tuned LLMs. This extends its practical applicability and offers a cost-efficient, scalable solution for treatment effect estimation under hidden confounding in real-world settings.

### E.2   IMPACT OF TEMPERATURE ON CONFOUNDER QUALITY

**Experimental Setup.** To assess how the temperature coefficient $\lambda$ affects the quality of generated confounders, we conduct a controlled experiment using two open-source LLMs: **LLaMA 3–8B** (**ProCI-La**) and **Qwen2.5–7B** (**ProCI-Qw**). We vary $\lambda$ in the range $\{0.1, 0.3, 0.5, 0.7, 0.9\}$, and for each value, use the respective LLM to generate confounders following the standard ProCI framework. To evaluate the informativeness of the generated variables, we compute their conditional mutual information (CMI) with the treatment and outcome, conditioned on observed covariates: $I(U, Y \mid X)$ and $I(U, T \mid X)$. We also include a **Random** baseline that generates synthetic variables from uniform or Gaussian noise independent of the data. This helps isolate the contribution of semantically meaningful generation from LLMs.

**Results.** Figure 6 shows the CMI values $I(U, Y \mid X)$ and $I(U, T \mid X)$ across different temperature values. Each x-axis tick corresponds to a specific temperature coefficient $\lambda$, and we compare three methods: Random, ProCI-La, and ProCI-Qw.

- **ProCI-La and ProCI-Qw significantly outperform the Random baseline at all temperatures**, with CMI values often one to two orders of magnitude higher, validating that LLM-generated confounders encode semantically relevant information about treatment and outcome.

- **The impact of temperature varies by method and metric.** For $I(U, Y \mid X)$, ProCI-La achieves the highest value at $\lambda = 0.5$, while ProCI-Qw peaks at $\lambda = 0.9$. For $I(U, T \mid X)$, ProCI-La performs best at the lowest temperature ($\lambda = 0.1$), whereas ProCI-Qw shows more stable performance across mid-to-high temperatures.

- **ProCI-La generally outperforms ProCI-Qw in capturing outcome-relevant information**, especially at lower temperatures. In contrast, ProCI-Qw sometimes surpasses ProCI-La on treatment-related informativeness ($I(U, T \mid X)$), suggesting complementary strengths between the LLMs.

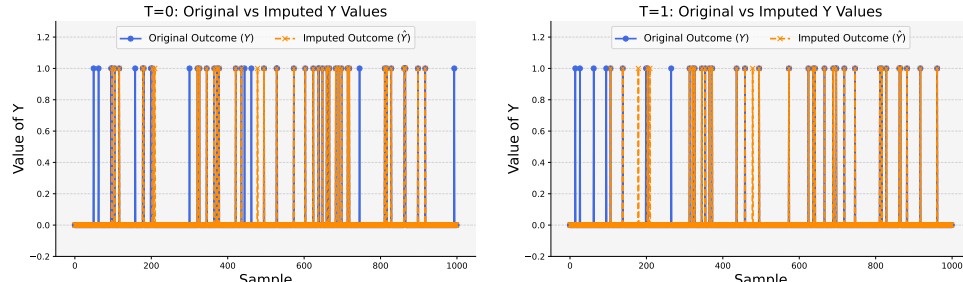

Figure 6: **Comparison between the original and imputed outcomes ($Y$ and $\hat{Y}$) for treatment $T = 0$ and $T = 1$, respectively.** The blue line represents the original $Y$ values, while the orange dashed line shows the simulated imputed outcomes $\hat{Y}$. The plot illustrates the effectiveness of the imputation process in approximating the original outcomes.

Table 5: Semantic comparison between removed covariates and imputed confounders. The number of ✓ symbols indicates the level of semantic matching: three checkmarks for a full match, two for a strong relation, and one for a weaker relation.

| Covariate | Scope | Confounder Generated by ProCI | | Matching |
|---|---|---|---|---|
| | | **Name** | **Scope** | |
| Age | $[17, 55]$ | Age | $[20, 50]$ | ✓✓✓ |
| Education years | $[3, 16]$ | Education background | {Bachelor, Master, ...} | ✓✓ |
| Married | $\{0, 1\}$ | Marital status | {Single, Married, Divorced} | ✓✓ |
| Hispanic | $\{0, 1\}$ | Living country | {USA, UK, Italy, ...} | ✓ |
| Previous earning | $[0, 37431]$ | Average Income per year | $[0, 120000]$ | ✓✓ |

These findings reinforce the benefit of using temperature tuning to control the diversity and informativeness of generated confounders. Moderate values ($\lambda \in [0.5, 0.7]$) typically offer the best trade-off, with performance degrading slightly at the extremes.

### E.3 QUALITY OF IMPUTED POTENTIAL OUTCOME

**Experimental Setup.** The Twins dataset contains real potential outcomes, where the outcome variable indicates whether a baby survived or died one year after birth, with 0 representing survival and 1 representing death. In this experiment, we test the quality of the potential outcomes imputed by our ProCI framework using GPT-4o, comparing them with the real potential outcomes in the dataset. We respectively test on two treatment groups: $T = 1$ and $T = 0$, where $T = 1$ represents babies with higher birth weight and $T = 0$ represents babies with lower birth weight. We aim to assess how well our framework can approximate the real potential outcomes for these two treatment conditions. Due to space limitations, we only report the results for the first 1000 samples.

**Results.** The results demonstrate that our ProCI framework effectively approximates the real outcome curve, especially in scenarios where the potential outcome is rare (e.g., $Y = 1$). This highlights the ability of ProCI to leverage its world knowledge for counterfactual estimation, successfully imputing potential outcomes even when direct observation is impossible.

### E.4 SEMANTIC COMPARISON BETWEEN REMOVED COVARIATE AND IMPUTED CONFOUNDER

**Experimental Setup.** In this experiment, we aim to evaluate how well our ProCI framework can restore the covariates that have been removed from the dataset, thereby demonstrating its ability to capture the underlying relationships within the data. Specifically, we remove one covariate at a time from the Jobs dataset and then use our framework to generate confounders, including both the semantic information and the value range of the imputed confounder. We then compare the

generated confounder with the removed covariate, categorizing the results into three groups: fully matched, strongly related, and weak relation.

**Results.** The results show that, semantically, while the ProCI framework's ability to restore the removed covariates varies across different covariates, it successfully generates confounders that are quite related to the removed covariates in most cases. However, in terms of value ranges, there is a noticeable difference between the range of values generated by our model and the original covariates. For instance, when the covariate "education years" is removed, it frequently reappears as a generated confounder, such as "education background". This is a reasonable and expected outcome, as both "education years" and "education background" are closely related in terms of semantic meaning and likely share similar underlying concepts. Such results suggest that the model effectively captures latent relationships, even if it doesn't always perfectly match the exact value ranges of the original covariates.

## F PROMPT TEMPLATES

In this section, we present the detailed prompt templates used in the proposed ProCI framework. These prompts correspond to the four key components of our method: variable generation ($\mathcal{P}_{\text{var}}(X, Y, T)$ in Eq. (4)), distribution type inference ($\mathcal{P}_{\text{dist}}(X, Y, T, \hat{U})$ in Eq. (5)), parameter estimation ($\mathcal{P}_{\text{param}}(x_i, t_i, y_i)$ in Eq. (6)), and counterfactual outcome imputation ($\mathcal{P}_{\text{out}}(x_i, u_i, y_i, t_i)$ in Eq. (7)). All corresponding equations are provided in the main paper, and this appendix serves to elaborate on the concrete prompt implementations used for each component.

### F.1 PREFIX PROMPT

The prefix prompt provides essential contextual information about the observational dataset, including a brief overview and detailed descriptions of the treatment, outcome, and confounding variables. This prompt serves as a foundation and should be included at the beginning of all subsequent prompts to ensure that the LLM is aware of the data background and variable semantics.

---

**Prefix prompt: Dataset Introduction**

**Inputs:** The dataset name $\mathcal{D}_{\text{name}}$ with a brief introduction $\mathcal{D}_{\text{intro}}$; variable names for confounders $X_{\text{name}}$, treatment $T_{\text{name}}$, and outcome $Y_{\text{name}}$; and their corresponding descriptions as provided by the original dataset: $X_{\text{desc}}$, $T_{\text{desc}}$, and $Y_{\text{desc}}$.

- - - - - - - - - - - - - - - - - - - - - - - - - - - - - - - - - - - - - - - - - -

**Prompt:**

Brief introduction of the $\{\mathcal{D}_{\text{name}}\}$ dataset: $\{\mathcal{D}_{\text{intro}}\}$

This observational dataset contains:

(1) Treatment — $\{T_{\text{name}}\}$: $\{T_{\text{desc}}\}$

(2) Outcome — $\{Y_{\text{name}}\}$: $\{Y_{\text{desc}}\}$

(3) Confounders — $\{X_{\text{name}}\}$: $\{X_{\text{desc}}\}$

---

### F.2 VARIABLE GENERATION

In this prompt, we mainly utilize the name and description of variables $X, T$ and $Y$ in the observational dataset to infer a new confounder $\hat{U}$.

---

**$\mathcal{P}_{\text{var}}(X, Y, T)$: Generating new confounder $\hat{U}$**

**Inputs:** Prefix Prompt

**Outputs:** Confounder name $\hat{U}_{\text{name}}$, a brief explanation $\hat{U}_{\text{exp}}$

- - - - - - - - - - - - - - - - - - - - - - - - - - - - - - - - - - - - - - - -

**Prompts:**

{Prefix prompt}

Based on your WORLD KNOWLEDGE, please propose one additional confounder which BOTH affects the treatment and outcome.

Make sure that the proposed confounder has a DIFFERENT MEANING compared to existing confounders.

For this proposed confounder, please provide:

(1) A clear name for the confounder.

(2) A brief explanation of why it affects both treatment and outcome.

---

### F.3 DISTRIBUTION TYPE INFERENCE

After identifying the confounder variable, we leverage the commonsense knowledge embedded in LLMs to infer an appropriate distribution type for it. Instead of directly imputing its values using the LLM, which often leads to degenerate or collapsed outputs when applied to tabular data, we defer value imputation to a subsequent structured process.

---

**$\mathcal{P}_{\text{dist}}(X, Y, T, \hat{U})$: Inferring the distribution type of $\hat{U}$**

**Inputs:** Prefix prompt, name of generated variable $\hat{U}_{\text{name}}$

**Outputs:** Distribution type $\mathcal{F}_{\hat{U}}$

- - - - - - - - - - - - - - - - - - - - - - - - - - - - - - - - - - - - - - - -

**Prompts:**

{Prefix Prompt}

Based on your WORLD KNOWLEDGE, please provide the distribution type of confounder $\{\hat{U}_{\text{name}}\}$. For example:

(1) Continuous — e.g., Normal distribution

(2) Discrete — e.g., Multi-categorical distribution

(3) Binary — e.g., Bernoulli distribution

---

### F.4 PARAMETER ESTIMATION

Given the inferred distribution type $\mathcal{F}_{\hat{U}}$ of the confounder $\hat{U}$, the next step is to estimate the corresponding distribution parameters. As there are various possible distribution families, we illustrate the parameter estimation process using the normal distribution as an example.

---

**$\mathcal{P}_{\textbf{param}}(\mathcal{D}_X, \mathcal{D}_T, \mathcal{D}_Y)$: Generating the distribution parameter for each unit**

**Inputs:** Prefix prompt, the confounder name $\hat{U}_{\text{name}}$, the values of confounder $\mathcal{D}_X$, treatment $\mathcal{D}_T$ and outcome $\mathcal{D}_Y$

**Outputs:** Distribution parameter $\theta_i = \{\mu_i, \sigma_i\}$ for each individual $i$

- - - - - - - - - - - - - - - - - - - - - - - - - - - - - - - - - - - - - - - - - - - - -

**Prompts:**

{Prefix prompt}

The values of existing confounders, treatments, and outcomes are given by:

(1) Confounder Values: $\{\mathcal{D}_X\}$

(2) Treatment Values: $\{\mathcal{D}_T\}$

(3) Outcome Values: $\{\mathcal{D}_Y\}$

For the confounder $\{\hat{U}_{\text{name}}\}$, please specify a normal distribution (mean and standard deviation) for each individual from which we can sample the confounder value.

---

To accommodate the token limitations of LLMs, this prompt is executed in a mini-batch manner. Once the distribution parameters (e.g., mean and standard deviation in the case of a normal distribution) are obtained, we sample concrete values of the confounder $\hat{U}$ from the personalized distribution for each instance.

## F.5 COUNTERFACTUAL OUTCOME IMPUTATION

To evaluate the effectiveness of the generated confounders, we assess whether the unconfoundedness assumption holds after incorporating them. Since LLMs implicitly encode a wide range of commonsense and domain-specific knowledge—including information related to potential hidden confounders—we utilize the LLM to impute counterfactual outcomes, $\hat{Y}^0$ and $\hat{Y}^1$. These counterfactuals are then used to perform an empirical test of the unconfoundedness assumption via conditional independence analysis.

---

**$\mathcal{P}_{\textbf{out}}(\mathcal{D}_X, \mathcal{D}_{\hat{U}}, \mathcal{D}_T, \mathcal{D}_Y)$: Imputing Counterfactual Outcomes**

**Inputs:** Prefix prompt, the values of confounder $\mathcal{D}_X$, imputed confounder $\mathcal{D}_{\hat{U}}$, treatment $\mathcal{D}_T$ and outcome $\mathcal{D}_Y$

**Outputs:** Counterfactual outcomes in $\hat{Y}^0$ and $\hat{Y}^1$

- - - - - - - - - - - - - - - - - - - - - - - - - - - - - - - - - - - - - - - - - - - - -

**Prompts:**

{Prefix prompt}

The values of existing confounders, treatments, and outcomes are given by:

(1) Confounders: $\{\mathcal{D}_X\}$

(2) Treatments: $\{\mathcal{D}_T\}$

(3) Outcomes: $\{\mathcal{D}_Y\}$

Based on the OBSERVED DATA and your WORLD KNOWLEDGE, please infer the values of the counterfactual outcome corresponding to the alternative value of treatment.

---

# G  CASE STUDY

In this section, we provide two running samples from Jobs dataset for both variable generation and value imputation.

## G.1  CASE ON VARIABLE GENERATION

---

**Generating New Variable {TRANSPORTATION ACCESS}**

**Prompts:**

Brief introduction of the {Jobs} dataset:

{The Jobs dataset is widely used in causal inference research for evaluating the performance of treatment effect estimation methods. It is constructed by combining experimental and observational data from the National Supported Work (NSW) demonstration and comparison group data (e.g., PSID or CPS).}

This observational dataset contains:

(1) Treatment — {Participation in Job Training}: {$T \in \{0, 1\}$ indicating whether the individual participated or did not participate in the job training program.}

(2) Outcome — {Employment Status}: {The individual's employment observed after the treatment decision. $Y = 1$ means employed, $Y = 0$ means not employed.}

(3) Confounders — {Age, Marriage, Previous Earning,...}: {Features affecting both the treatment and the outcome, such as age, education, prior income, etc.}

Based on your WORLD KNOWLEDGE, please propose one additional confounder which BOTH affects the treatment and outcome.

Make sure that the proposed confounder has a different meaning compared to existing confounders.

For this proposed confounder, please provide:

(1) A clear name for the confounder.

(2) A brief explanation of why it affects both treatment and outcome.

- - - - - - - - - - - - - - - - - - - - - - - - - - - - - - - - - - - - - - -

**Answers:**

(1) Variable Name: Transportation Access

(2) Explanation: Access to reliable transportation can influence both participation in the job training program (treatment) and subsequent employment (outcome). Individuals without transportation may be less likely to enroll in or attend the program due to logistical barriers. Similarly, lack of transportation can hinder job search efforts and commuting to workplaces, reducing the likelihood of employment.

---

## G.2 CASE ON VALUE IMPUTATION

---

**Inferring Distribution Type for New Variable {TRANSPORTATION ACCESS}**

**Prompts:**

Brief introduction of the {Jobs} dataset:

{The Jobs dataset is widely used in causal inference research for evaluating the performance of treatment effect estimation methods. It is constructed by combining experimental and observational data from the National Supported Work (NSW) demonstration and comparison group data (e.g., PSID or CPS).}

This observational dataset contains:

(1) Treatment — {Participation in Job Training}: {$T \in \{0, 1\}$ indicating whether the individual participated or did not participate in the job training program.}

(2) Outcome — {Employment Status}: {The individual's employment observed after the treatment decision. $Y = 1$ means employed, $Y = 0$ means not employed.}

(3) Confounders — {Age, Marriage, Previous Earning,...}: {Features affecting both the treatment and the outcome, such as age, education, prior income, etc.}

Based on your WORLD KNOWLEDGE, please provide the distribution type of confounder {Transportation Access}. For example:

(1) Continuous — e.g., Normally distribution

(2) Discrete — e.g., Multi-categorical distribution

(3) Binary — e.g., Bernoulli distribution

---

**Answers:**

(1) Distribution Type: Binary

(2) Value Description: 0 indicates the individual lacks reliable transportation (e.g., no personal vehicle or inadequate public transit access), and 1 indicates they have reliable transportation.

---

**Estimating Values for New Binary Confounder {TRANSPORTATION ACCESS}**

**Prompts:**

Brief introduction of the {Jobs} dataset:

{The Jobs dataset is widely used in causal inference research for evaluating the performance of treatment effect estimation methods. It is constructed by combining experimental and observational data from the National Supported Work (NSW) demonstration and comparison group data (e.g., PSID or CPS).}

This observational dataset contains:

(1) Treatment — {Participation in Job Training}: {$T \in \{0, 1\}$ indicating whether the individual participated or did not participate in the job training program.}

(2) Outcome — {Employment Status}: {The individual's employment observed after the treatment decision. $Y = 1$ means employed, $Y = 0$ means not employed.}

(3) Confounders — {Age, Education Years, ..., Previous Earnings}: {Features affecting both the treatment and the outcome, such as age, education, prior income, etc.}

The values of existing confounders, treatments, and outcomes are given by:

(1) Confounder Values: {[47,12,...,0], ..., [40, 8, ..., 0]}

(2) Treatment Values: {0,0,1,0,...,1}

(3) Outcome Values: {1,0,0,1,...,0}

For the new confounder {Transportation Access}, please specify a binary value for each individual.

- - - - - - - - - - - - - - - - - - - - - - - - - - - - - - - - - - - - - -

**Answers:**

(1) Values: [0,0,0,...,1]

(2) Summary: The first 82 individuals (previous earning = 0) lack reliable transportation (Transportation Access = 0), while the remaining 68 (previous earning $> 0$) have access (Transportation Access = 1). This aligns with prior earnings as a proxy for transportation availability, creating a plausible confounder structure.

## H  BROADER IMPACTS

This work introduces ProCI, a framework that uses LLMs to help mitigate hidden confounding when estimating treatment effects from observational data. By combining structured and unstructured data with the general knowledge and reasoning abilities of LLMs, ProCI offers a new and useful way to improve causal inference. In practice, the ProCI framework can improve decision-making in areas where running controlled experiments is difficult or unethical, such as healthcare, social programs, and economic policies. It can help uncover hidden confounders that affect treatment and outcomes, leading to fairer and more informed decisions—especially in places with limited resources. Also, by reducing the need for expert-designed tools or domain knowledge, ProCI makes causal analysis easier and more available to a wider group of researchers and practitioners.

## I  LIMITATIONS

While our proposed method consistently outperforms base models across benchmarks, it still exhibits several limitations that warrant further investigation:

- Our evaluation is conducted on two widely-used observational benchmarks—Twins and Jobs. To better assess the generalizability of the ProCI framework, future work should explore a broader range of real-world and domain-specific datasets.
- As noted in Theorem 1, the empirical unconfoundedness test using Kernel Conditional Independence Test (KCIT) on imputed counterfactuals approximates the true test only when the sample size is sufficiently large. More robust or distribution-free statistical tests may be needed to relax this assumption in smaller datasets.

- While our study extends confounder generation to include four distinct LLMs—GPT-4o, DeepSeek-R1, LLaMA 3–8B, and Qwen2.5–7B—these models primarily represent instruction-tuned decoders. Future work should further examine ProCI's applicability across a wider range of architectures, such as multilingual models, encoder-decoder frameworks, or smaller-scale LLMs, to comprehensively assess its robustness and scalability.

