# OpenReview forum: "Mitigating Hidden Confounding by Progressive Confounder Imputation via Large Language Models"
_ICLR.cc/2026/Conference — ICLR 2026 Conference Withdrawn Submission_

### Official Review · Reviewer_5wjc · 2025-10-27

**Soundness:** 1
**Presentation:** 3
**Contribution:** 2
**Rating:** 2
**Confidence:** 3

**Summary:**

The paper propose a complete framework for _hidden confounder_ imputation, using Large Language Models, called ProCI.
The workflow consists of several steps: 1) extract the name and the description of the hidden confounder, 2) extract the distribution of the hidden confounder, 3) compute the values of the distribution of the hidden confounder for each datapoint, based on its description and the values of the covariates, the treatment and the outcome, 4) compute a conditional independence test, to assess conditional unchangeability
$(Y^1, Y^0) \perp T | X, \hat{U}$, and 5) repeat 1) to 4) until the independence condition is met. In addition, the authors validate these imputations evaluating the performance of causal inference methods on benchmarking small structured data (Twins and Jobs), concluding that ProCI provides better metrics than using estimators that do not account fo hidden confounders. Other questions, regarding robustness, mutual information between $T$ and $U$, and the components of the LLMs needed (distributed reasoning, progressive imputation, independence tests) have been carried out. Those experiments report that LLM imputed confounders contain information about the treatment that is not present in the covariates, $X$, in both datasets; that all components are important for imputation, specially distributed reasoning; and that the framework is robust to variable removal until some point.
In addition to the practical considerations, the authors provide theoretical assymptotic guarantees of convergence of the conditional independence test, which states that the conditional exchangeability is satisfied with the estimated $U$, if the real $U$ satisfies it, given an infinite number of samples. However, this guarantee depends on untestable assumptions, i.e., for that theorem to hold, the LLM must capture the distribution not only of the hidden confounders, but also of the unobserved potential outcomes. This assumption is not testable in practice. Consequently, in general there are not theoretical guarantees of operation, and the experiments, although systematic, are scarce and should reflect more complex real-world problems.

**Strengths:**

- The paper clearly defines the problem, so the motivation is fair and the proposed workflow can be easily understood. The procedure of progressive imputation is conceptually consistent, and the stopping criterion with conditional independence tests provides an interpretable mechanism for validating the confounders. The use of distributional reasoning mitigates mode collapse and it is experimentaly ablated. The method is model-agnostic, making it adaptable to multiple models.
 - The paper proposes to leverage the knowledge of huge LLMs to extract information about confounders. It is sound that the external world knowledge of the LLM can name plausible hidden confounders.
 - The research questions discussed in the _experiments_ section are clearly stated and they do not limit only to accuracy performance in causal effect estimation. The experiments are systematic and report improvements over all models.

**Weaknesses:**

I have several important concerns:
 - In the theorem 1, and associated Lemmas 1 and 2, the authors provide asymptotic guarantees of KCIT (kernel conditional independence test). However, this implies that the distribution of both hidden confounders ($\hat{U}$) and unobserved potential outcomes ($\hat{Y}^{1-T}$) is well captured by the LLM. However, how can this be guaranteed in practice? The authors should adress at least discussion on this assumptions, and recall that this _guarantee_ depends on an untestable assumption. In addition, we should note that both $\hat{Y}^{1-T}$ and $\hat{U}$ come from and LLM: if the LLM hallucinate in both variables to satisfy independence by construction, while the causal sufficiency would not be covered in reality.

- When using LLMs, and its so-mentioned world knowledge, the practitioner does not have any guarantee about the prediction that the LLM yields. While it is plausible to extract the names of the confounders, I have more concerns about their imputation. In `line 277`, the authors claim that LLMs can be viewed as unbiased counterfactual estimators under unconfoundedness. However, there are not specific reasons or rigurous proofs to validate that claim. On the contrary, it is well known that LLMs are not, in general, reliable tabular regressors [1]. In addition, the risk of hallucination when imputing many numerical instances increases, and that problem is not adressed in the paper.

- Conditional independence information comes from a cyclic relationship. That is, $Y$ and $T$ are included in the prompt to impute $\hat{U}$, which makes the imputed variable a post-treatment variable. This assumption may violate the causal ordering. The only reason because we should believe that $\hat{U}$ is a confounder instead a collider, a mediator, or other kind of variable is because the LLMs told that, but there is not proposed way to verify that. Probably ensemble several LLMs could help to minimize the risk.

- The authors should provide reasons because an LLM should think causally when imputing $U$ or $Y$. In addition, they should discuss about transportability problems; i.e., what happens if the data seen by the LLM does not match with the distribution of the target data?

- Although the experiments are systematic, the scope is limited. These benchmarks are small, structured, and usually used when _unconfoundedness_ hold. Then, extended evaluation with real-world data, or large scale unstructured datasets (e.g. EHRs) may help to understand the operation of the framework.

Minor concerns:
  - I have one notation problem: in Eq. 3, $\hat{U}$ is defined as _the name of a random variable_ (which is itself a random variable), while later, e.g. Eq. 8, the same letter is used to denote _a random variable_. This notation can be misleading, and I recommend to check the meanings of those variables.

- In Lemma 2, it is not well explained what LC means, and it is probably needed to understand the conditions.

 -The reproducibility is fragile because it depends on prompts, temperatures and specific LLMs. Do the authors have any idea of how to reduce this variability?

- In literature review, probably information about proximal inference is required for completeness.

> summary

Overall, I find this paper promising but not causally sound. The fact that $\hat{U}$ builds a sufficient adjustment set is not verifiable, and the KCIT condition not only relies on also unverifiable assumptions, but also is not a proof of $\hat{U}$ to build a sufficient adjustment set. Although the experiment are systematic and well explained, since the LLMs are huge black-boxes without explicit causal knowledge, a more robust evaluation on real-world unstructured data may help to adopt this type of workflow in practice. Since there are not theoretic testable guarantees of operation, and the experiments in real-world complex data have not been addressed, I cannot recommend the acceptation of this paper in its current state. However, I encourage the authors to  keep working in this line, since it may open many intriguing venues.


[1] Hegselmann, S., Buendia, A., Lang, H., Agrawal, M., Jiang, X., & Sontag, D. (2023, April). Tabllm: Few-shot classification of tabular data with large language models. In International conference on artificial intelligence and statistics (pp. 5549-5581). PMLR.

**Questions:**

The most of the questions I had are in the previous sections, but I am curious about two questions.

- If the authors considered that the prediction of $Y$ by an LLM is good enough to run a KCIT, why not directly estimate the potential outcome with an LLM, instead of a causal inference model? Probably an experiment comparing TARNet, CFRNet, etc., with LLMs regressors for potential outcomes may help to understand the claim 'LLMs [...] can be viewed as approximately unbiased counterfactual estimators...'

- When evaluating robustness by removing variables in well-known datasets, as Jobs, is it possible that the LLM has access to the true values of the removed variables in the dataset?

---

### Official Review · Reviewer_mkmQ · 2025-10-31

**Soundness:** 3
**Presentation:** 4
**Contribution:** 3
**Rating:** 6
**Confidence:** 4

**Summary:**

The authors propose ProCI (Progressive Confounder Imputation), a novel framework that leverages large language models (LLMs) to iteratively generate, impute, and validate hidden confounders. ProCI exploits the LLMs’ semantic reasoning and embedded world knowledge to discover plausible confounders, imputes their distributions rather than fixed values to improve robustness, and tests unconfoundedness via counterfactual reasoning and a KCIT-based independence test. Experiments on benchmark datasets (Twins and Jobs) show that ProCI substantially improves treatment effect estimation across multiple models (S-Learner, TARNet, CFR-Wass, ESCFR).

**Strengths:**

1. The paper is well-written and easy to follow despite the technical depth, with helpful figures and consistent terminology.

1. The idea that utilizing the world knowledge from LLM to impute unobserved confounders is novel and also interesting. This represents a creative and timely direction bridging causal inference and foundation models.

1. The proposed Unconfoundedness Validation is new, reasonable, and well aligned with the overall framework.

1. The experimental results are comprehensive, well supporting its claims.

Overall, despite the simplicity of the proposed idea, it proves to be both effective and insightful. I appreciate this balance between conceptual clarity and practical impact.

**Weaknesses:**

1. The framework’s performance and reliability may vary substantially with different LLMs. It would be helpful to discuss potential limitations or safeguards when weaker or domain-mismatched models are used.
1. While Theorem 1 provides some justification, a more detailed discussion of theoretical limitations or assumptions on the LLM’s implicit confounder knowledge would strengthen the rigor.
1. How is the significance level $\alpha$ chosen in the KCIT test? Does it affect the accuracy of CATE estimation?
1. In the Unconfoundedness Validation phase, how reliable are the generated counterfactual outcomes? Could they potentially serve as estimates for individual treatment effects?
1. The iterative prompting and KCIT testing could be expensive in large-scale scenarios; some empirical timing or efficiency analysis would be valuable.

I am happy to further increase my score if the above questions and clarifications can be well addressed.

**Questions:**

see above weaknesses

---

### Official Review · Reviewer_P59B · 2025-10-31

**Soundness:** 1
**Presentation:** 3
**Contribution:** 3
**Rating:** 2
**Confidence:** 4

**Summary:**

This paper proposes ProCI, a progressive confounder imputation framework that uses LLMs to generate candidate latent confounders, impute their values via distribution-aware prompting, and validate sufficiency through an empirical conditional independence check based on LLM-imputed counterfactuals. The method alternates between confounder generation and a kernel conditional independence test until the independence criterion is met, after which standard CATE estimators are trained on the augmented data. Experiments on Twins and Jobs data with several base estimators and LLMs report improvements over baselines, with an asymptotic argument justifying the use of imputed outcomes in the independence test.

**Strengths:**

The paper addresses an important pain point in observational causal inference and articulates a clear pipeline. Progressive confounder generation and distributional reasoning are interesting design choices that seem to empirically mitigate collapse in direct value imputation and appear to yield semantically coherent variables. The validation procedure is easy to integrate with existing estimators and the method is model-agnostic in practice. The experimental section is comprehensive for the chosen benchmarks, spans several estimator families, and probes the sensitivity of the approach to LLM choice and temperature. The submission is well written, and the core idea is original and, indeed, provocative, which could help stimulate further research in the area.

**Weaknesses:**

- The abstract and positioning somewhat overstate the novelty relative to the broader literature on handling unobserved confounding. Sensitivity analysis, instrumental-variable and front-door strategies, and designs that exploit temporal or quasi-experimental variation represent long-standing alternatives; I would consider tightening the claims accordingly.

- More consequentially, the empirical “unconfoundedness validation” is not a test of unconfoundedness in the data-generating process but a test on outcomes that the same class of LLMs has imputed. Using one model family to propose U, to impute Y(0) and Y(1), and to decide when the process stops risks a self-fulfilling loop that can pass independence by construction, or have poorly understood biases. This circularity, together with the fact that hidden confounders are unobserved by definition, makes it difficult to interpret a KCIT pass as evidence that bias has been removed rather than absorbed or masked by the imputation step. The paper would benefit from guardrails, such as using disjoint models that the generator and validator do not share parameters or prompts. In general, claims that unconfoundedness has been obtained are speculative; unobserved confounders remain unobserved, so we cannot directly assess performance against them.

- The LLM being used in confounder imputation might contain post-treatment information. This could generate post-treatment bias and harm inference.

- The LLM being used in inference might have knowledge of the datasets being used, and therefore information leakage could affect performance evaluations, with poorly understood effects.

- Given the just-outlined comments, I believe the paper would benefit from a complete re-frame where the proposed method is framed as a possibly harmful, possibly beneficial proof of concept that requires further research to validate.

- The method also appears to rely on informative variable names and human-readable descriptions when prompting for new confounders. Many real tabular datasets use terse, idiosyncratic codes, and clinical or enterprise schemas frequently obscure semantics. It is unclear how robust ProCI is under abbreviated or synthetic schemas or multilingual column names; stress tests in such settings would strengthen the case for generality.

- The implementation details seems to suggest all models are built in PyTorch yet GPT-4–series models are proprietary services; the paper should clarify the inference pathway (distinguish which models are built vs. called via API).

**Questions:**

How is leakage prevented between the confounder-generation prompts and the validation prompts?

How does ProCI behave when column names are anonymized or randomly permuted?

Are there cases where the generated confounders were found ex post to be unhelpful or harmful, and what heuristics or diagnostics are recommended for screening them?

**Details Of Ethics Concerns:**

None observed.

---

### Note · Authors · 2025-11-23

I have read and agree with the venue's withdrawal policy on behalf of myself and my co-authors.